# ENHANCING DIFFUSION MODELS TOWARDS UNIFIED GENERATIVE AND DISCRIMINATIVE LEARNING

## ABSTRACT

While diffusion models excel at image synthesis, useful representations have been shown to emerge from generative pre-training, suggesting a path towards unified generative and discriminative learning. However, suboptimal semantic flow within current architectures can hinder this potential: features encoding the richest high-level semantics are underutilized and diluted when propagating through decoding layers, impeding the formation of an explicit semantic bottleneck layer. To address this, we introduce *self-conditioning*, a lightweight mechanism that reshapes the model's layer-wise semantic hierarchy *without external guidance*. By aggregating and rerouting intermediate features to guide subsequent decoding layers, our method concentrates more high-level semantics, concurrently strengthening global generative guidance and forming more discriminative representations. This simple approach yields a dual-improvement trend across pixel-space UNet, UViT and latent-space DiT models with minimal overhead. Crucially, it creates an architectural semantic bridge that propagates discriminative improvements into generation and accommodates further techniques such as contrastive *self-distillation*. Experiments show that our enhanced models, especially self-conditioned DiT, are powerful dual learners that yield strong and transferable representations on image and dense classification tasks, surpassing various generative self-supervised models in linear probing while also improving or maintaining high generation quality.

## 1 INTRODUCTION

Diffusion models (Ho et al., 2020) have recently emerged as one of the most powerful and popular techniques in generative AI, renowned for their ability to synthesize photorealistic visual data. These models have demonstrated remarkable versatility and high performance across a spectrum of tasks, such as class-conditional generation (Peebles & Xie, 2023; Karras et al., 2024), text-to-image synthesis (Rombach et al., 2022; Esser et al., 2024) and image editing (Mokady et al., 2023; Brooks et al., 2023), along with flexible customization options (Ruiz et al., 2023; Zhang & Agrawala, 2023).

Meanwhile, interest has grown in repurposing pre-trained diffusion models for discriminative tasks. Studies have shown that the intermediate representations (Baranchuk et al., 2022; Xiang et al., 2023) are effective for downstream tasks, particularly dense prediction like segmentation (Xu et al., 2023), depth estimation (Zhao et al., 2023) and keypoint detection (Xu et al., 2025). These findings highlight diffusion's potential as **unified** generative-and-discriminative learners, which gain a deep understanding of data through generative pre-training, as in language models (Chen et al., 2020a).

Despite these advances, diffusion models still face challenges in representation learning: achieving optimal performance often necessitates specialized methods for feature extraction (Meng et al., 2024), distillation (Li et al., 2023), or design of dedicated decoders (Zhao et al., 2023). Moreover, while adept at capturing local semantics crucial for dense prediction, their global feature quality often underperforms modern self-supervised learners, especially in image-level tasks such as linear classification (Hudson et al., 2024; Chen et al., 2025). This discrepancy underscores a limitation in their ability to *condense high-level semantics into a compact, discriminative feature*.

Unlike paradigms imposing an explicit information bottleneck, such as view alignment in contrastive methods (Wang & Isola, 2020) or asymmetric encoder-decoder design in masked autoencoders (He et al., 2022), diffusion models distribute semantic information across all layers, with each offering representations of varying granularity (Baranchuk et al., 2022). While this dispersion is an emergent

and natural outcome of generative pre-training, it poses a fundamental challenge for representation learning, as no single layer is explicitly designed as a semantic bottleneck (Hudson et al., 2024).

While diagnostic studies such as DDAE (Xiang et al., 2023) have revealed a layer-wise hierarchy, where rich semantics emerge in middle-to-late layers, they are limited to post-hoc evaluation of pre-trained models. Moving from diagnosis to intervention, we posit that *the most semantically rich features are routed inefficiently and underutilized when propagating through decoding layers*. Such suboptimal information flow not only dilutes the guidance signals for

Table 1: **Trade-offs and gains in generation and representation.** We enhance standard diffusion models in both domains, without requiring major framework overhauls or external knowledge.

| | Generative Modeling | | Representation Learning | |
|---|---|---|---|---|
| | Standard, generalizable | Sample quality++ | Self-supervised, no extra encoder | Feature quality++ |
| l-DAE (Chen et al., 2025) | ✗ | ✗ | ✓ | ✓ |
| SODA (Hudson et al., 2024) | ✗ | ✓ | ✓ | ✓ |
| RCG (Li et al., 2024) | ✓ | ✓ | ✗ | ✗ |
| REPA (Yu et al., 2025) | ✓ | ✓ | ✗ | ✓ |
| DDAE (Xiang et al., 2023) | ✓ | ✗ | ✓ | ✗ |
| **DDAE++ (Ours)** | ✓ | ✓ | ✓ | ✓ |

synthesizing global structures, but also impedes the formation of a more explicit bottleneck layer. To resolve this, we introduce **self-conditioning**, a mechanism to structurally reshape model's intrinsic semantic hierarchy by feature rerouting, resulting in more concentrated representations (Fig. 5, 7).

Our high-level principle is simple: aggregate a rich semantic feature from an intermediate layer, and use it to condition subsequent decoding layers. To implement this efficiently, our strategy is to *reuse* the inherent conditioning pathways already present in diffusion backbones, allowing for tailored yet minimal modifications: for UNet (Dhariwal & Nichol, 2021) and DiT (Peebles & Xie, 2023), we leverage their adaptive normalizations by injecting a global pooled feature vector after a time-adaptive transformation (Fig. 1); for in-context conditioning UViT (Bao et al., 2023a), we employ a new summary token to automatically aggregate features and condition patch tokens via self-attention (Fig. 2). This flexible reuse strategy makes our approach a plug-and-play enhancement with minimal computational overhead, and demonstrates a broad applicability across diverse architectures.

The immediate outcome of self-conditioning is a dual benefit: improved sample and feature quality. More importantly, it forges an **architectural semantic bridge** that connects a discriminative feature to its generative decoding path. This bridge unlocks further potentials: dedicated representation learning methods, such as contrastive learning (Chen et al., 2021), can now be integrated to directly refine the concentrated feature. In turn, this enhanced feature is fed back through the bridge, providing stronger semantic guidance for generation. Our full framework, DDAE++, combines these synergistic components with self-conditioning to further amplify gains in both domains.

Extensive experiments demonstrate a generalizable trend, where accuracy is significantly boosted in most cases while FID is either improved or remains on par with strong diffusion baselines, in contrast to prior works that often sacrifice generative abilities (Hudson et al., 2024; Chen et al., 2025). Particularly noteworthy are UViT and DiT, which, with our enhancements, exhibit exceptional potential for representation learning, surpassing various self-supervised models. Crucially, the most substantial gains from discriminative techniques are observed when paired with self-conditioning, highlighting the synergy our approach facilitates between generative and discriminative paradigms.

In summary, our key contributions are:

- **Conceptual**: We identify an underexplored limitation in diffusion architectures, *i.e.*, suboptimal semantic flow, and show that correcting it yields coupled gains in generation and representation.

- **Methodological**: We propose self-conditioning, a lightweight mechanism that repurposes conditioning pathways to structurally consolidate semantics and synergize with discriminative methods.

- **Empirical**: Our approach is broadly effective across multiple models, backbones, and datasets. Notably, our enhanced DiT exhibits favorable scaling potential in discriminative performance.

## 2 RELATED WORK

**Self-supervised learning (SSL)** has established two dominant paradigms with distinct properties. Contrastive learning (CL) distills image-level semantics into a compact feature, pulling augmented views of an image closer. This instance discrimination (Wu et al., 2018) makes it highly effective for linear evaluation (Grill et al., 2020; Chen et al., 2021). Masked image modeling (MIM), conversely, learns by reconstructing corrupted inputs, akin to denoising autoencoding (Vincent et al., 2008). While this generative process preserves rich, transferable information, the resulting features are less

discriminative before fine-tuning (Bao et al., 2022b; He et al., 2022). As a form of denoising autoencoders (Xiang et al., 2023; Chen et al., 2025), diffusion naturally inherits a similar trade-off. Our work operates at the intersection of these paradigms to address it. We first enhance semantic aggregation and utilization via an architectural modification, and then integrate a contrastive framework, aligning with hybrid methods that unify CL and MIM (Zhou et al., 2022; Huang et al., 2023).

**Semantically enhanced diffusion models.** Unconditional models often lag behind conditional ones in generation quality (Bao et al., 2022a). To bridge this gap, some studies leverage vision foundation models to provide extra guidance, which can be in the form of pseudo-labels derived from clustering (Hu et al., 2023) or direct feature injection (Li et al., 2024). However, reliance on external signals limits the model's self-contained nature and compromises the flexibility of native unconditional applications (*e.g.*, image translation (Su et al., 2023) and domain adaptation (Liu et al., 2023)). Recent methods like REPA (Yu et al., 2025) use DINOv2 (Oquab et al., 2024) feature as a distillation target to regularize the output, rather than as an input condition. In contrast, our enhancement is self-contained. We posit that powerful semantic cues for guidance already exist within a model's feature hierarchy but remain underutilized. By improving the *internal* information flow, our approach fully preserves the original flexibility, and is conceptually orthogonal to *external* methods like REPA.

## 3 BACKGROUND

**Diffusion models.** Diffusion (Ho et al., 2020; Karras et al., 2022) and flow-based (Lipman et al., 2023; Liu et al., 2023) models are state-of-the-art generative models with strong theoretical connections (Esser et al., 2024). These models, hereinafter collectively referred to as *diffusion models*, construct paths that progressively transform data into noise via a time-forward process over $t \in [0, T]$:

$$x_t = \alpha_t x_0 + \sigma_t \epsilon \quad \text{where } \epsilon \sim \mathcal{N}(0, I), \tag{1}$$

with schedules for $\alpha_t, \sigma_t$ such that $t = 0$ corresponds to $p_{data}$ and $t = T$ approximates $\mathcal{N}(0, I)$. To reverse this, an ordinary differential equation (ODE) is typically formulated (Song et al., 2021b):

$$\mathrm{d}x_t = v_\theta(x_t, t)\mathrm{d}t, \tag{2}$$

where velocity $\mathrm{d}x_t/\mathrm{d}t$ is parameterized by a time-conditioned network $v_\theta$, which, once trained, enables ODE solvers to generate data. The estimation of $v_\theta$ is closely related to denoising autoencoding and score matching (Song et al., 2021b), allowing the training objective to be reparameterized into flexible forms, typically including training the network to predict the added noise $\epsilon$, the clean data $x_0$, or the velocity. For simplicity, we refer to any such objective as *diffusion loss*, denoted by $\mathcal{L}_{\text{diff}}$.

Different formulations of diffusion models may vary in noise schedule, training objective and ODE solver. In this paper, we examine three representative ones: DDPM (Ho et al., 2020), EDM (Karras et al., 2022) and Rectified Flow (RF) (Liu et al., 2023), with detailed overview provided in Appx. A.

**Backbones for diffusion models.** Early works like DDPM adapt a UNet (Ronneberger et al., 2015) architecture, where time $t$ is specified through a global conditioning pathway, by injecting sinusoidal embeddings (Vaswani et al., 2017) of $t$ into each block. We refer to this basic version as `ddpm`[1]. DDPM++ (Song et al., 2021b) further builds upon this, enhancing its capacity by doubling depth and employing BigGAN-style blocks (Brock et al., 2018). We refer to this scaled-up variant as `ddpmpp`.

Recently, ViT-based backbones have demonstrated better scalability (Bao et al., 2023b; Esser et al., 2024). We examine two designs with distinct conditioning mechanisms: UViT (Bao et al., 2023a), which processes all inputs (including time and other conditions) as tokens in a transformer; and DiT (Peebles & Xie, 2023), which injects conditions via AdaLN, analogous to UNet's AdaGN (Dhariwal & Nichol, 2021). The architectural similarity of these backbones to those used in visual recognition (Dosovitskiy et al., 2021) also motivates our investigation into their representation learning potential.

**Motivation for a comprehensive baseline.** Most related studies focus on extracting features from one specific model (Baranchuk et al., 2022; Yang & Wang, 2023), with few efforts to identify which models (and what factors) lead to better representations. While DDAE first suggested a link between generation and recognition by comparing DDPM and EDM, its analysis had limitations: the comparison lacked strict control over confounding factors (*e.g.*, backbone sizes, augmentations), and its scope did not cover modern advances like flow-based models or ViT-based backbones.

---

[1] We distinguish backbones (notations like `ddpm`) from models (DDPM), which can be combined flexibly.

(a) **UNet** with self-conditioning: an illustration.

```
# blockno    : the layer index between enc-dec
# t_embedder : a standalone timestep embedder
# proj       : a single nn.Linear layer

def dit_forward(x, t, c=None):
    x = x_embedder(x) + pos_embed        # (N, T, D)
    e = t_and_c_embedder(t, c)           # (N, D)
    for no, block in enumerate(dit_blocks):
        x = block(x, e)                  # (N, T, D)
        if no + 1 == blockno:
            global_feat = x.mean(dim=1)  # -->(N, D)
            e = e + t_embedder(t) * proj(global_feat)
    x = final_layer(x, e)
    return unpatchify(x)
```

(b) **DiT** with self-conditioning: PyTorch-like pseudocode.

Figure 1: **Self-conditioning applied to backbones based on adaptive normalization.** Originally, time $t$ (and optional condition $c$) are specified via a global conditioning pathway, where their embedding $e$ is injected into all layers. Here, we collect features from a specific layer by average pooling, and add them to the pathway of decoding layers, after being projected and time-adaptively scaled.

To establish a solid foundation for fair and unified comparison, and to validate our method's generalizability, we build a more comprehensive and controlled baseline. It spans multiple model formulations, backbones (of different types and sizes) and datasets, ensuring a rigorous demonstration of our method's benefits over its corresponding baseline across settings (Tab. 2).

## 4    APPROACH

Our DDAE++ framework enhances diffusion models by systematically addressing their suboptimal and underutilized representations. It is built upon three complementary perspectives: **(1)** The architectural foundation is our core contribution, *self-conditioning*, which better produces and utilizes the semantics, and creates a robust bridge between representation and generation pathways. The bridge is a critical enabler for two other perspectives, translating further improved semantics into improved generation. **(2)** From the data-space, we employ *non-leaky augmentations* to generally improve representation robustness. **(3)** From the objective-space, we introduce *contrastive self-distillation* to directly refine the newly formed bottleneck. Together, these architectural, data-space and objective-space components form a synergistic system for dual enhancement.

### 4.1    SELF-CONDITIONING

The essence of self-conditioning is to create a feedback loop: aggregating useful features from an intermediate layer and rerouting them to guide subsequent decoding layers. However, a conflict arises due to diffusion model's nature: effective representation often occurs in a narrow range of timesteps, while high-quality generation relies on the entire trajectory. The aggregated features, while valuable within a certain range, can be too noisy to provide beneficial guidance in others. Therefore, our implementations are designed to be **time-adaptive**, dynamically modulating the features. Specifically:

**For UNet and DiT,** our method re-injects features from an intermediate layer back into the decoder's conditioning pathway. As shown in Fig. 1a, we apply global average pooling to the encoded feature map of a designated layer to obtain a compact vector. It is then projected and modulated by a learned, time-dependent scaling factor before being added to the original condition embedding (pseudocode in Fig. 1b). This enriched embedding carrying global context is propagated to all subsequent layers.

Unlike works relying on FID-based hyperparameter tuning (Yu et al., 2025), we leverage training dynamics to determine the self-conditioning layer: inspired by Fig. 6, we posit that *faster convergence signals better information flow and higher improvement*. We thus conduct short, parallel training runs (*e.g.*, 20 epochs) on each mid-to-late layer (*e.g.*, 8-11 in DiT-B), and select the one with lowest loss. We validate in Sec. 5.4 that this early-stage loss is a strong predictor of final FID and accuracy.

**For UViT,** our approach automates the entire self-conditioning process, *i.e.*, semantic aggregation, time-adaptive modulation and decoding guidance, through an attention-native mechanism. Since all information, not only patch tokens but also the time token, is globally and continuously interacted, a

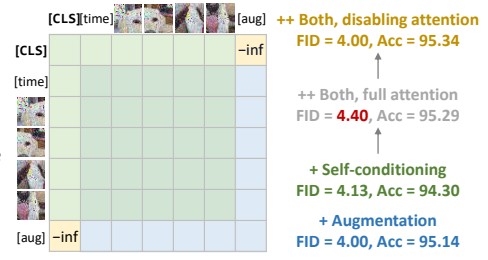

(a) **UViT** with `[CLS]` token, which automatically achieves time-adaptive semantic aggregation and decoding guidance.

(b) **Attention masks** when using non-leaky aug. It works well with either the `[CLS]` for self-conditioning, or the augment label for non-leaky aug. However, attention between them may be harmful when used together, while disabling this interaction resolves the issue.

Figure 2: **Self-conditioning applied to backbones based on in-context attention.** Based on the all-as-tokens design, we leverage an additional token to automatically interact with patch tokens and the time token, eliminating the need for manual feature selection, pooling, modulation or rerouting.

dedicated summary token is more natural than layer-wise pooling and vector scaling. We therefore introduce an additional learnable token, initialized randomly and conceptually empty, to dynamically aggregate and utilize global semantics as it propagates through the transformer (Fig. 2a).

We refer to this token as `[CLS]` due to its functional similarity to ViT's class token, and this design may also connect to register tokens (Darcet et al., 2024) and visual prompts (Jia et al., 2022). Note that despite the nomenclature, we *do not* apply any regularization to this token at this stage (learned solely through diffusion), in contrast to the class token used in supervised ViT training.

### 4.2 ADAPTING DISCRIMINATIVE LEARNING TECHNIQUES

**Non-leaky augmentations** serve a dual purpose in our framework: generally enhancing features and creating positive views for contrastive learning. To avoid harmful artifacts introduced by aggressive SSL transformations (Chen et al., 2020b), we use a non-leaky geometric pipeline (*e.g.*, translation, scaling, rotation) proposed by EDM (Karras et al., 2022). A 9-dim vector of transformation parameters is used to condition the model, preventing augmentation effects from leaking into generation (detailed in EDM's Tab. 6). However, this choice presents a trade-off, as these weaker transformations may yield less diverse views and limit contrastive effectiveness (Tian et al., 2020).

We also find that `[CLS]` and the augmentation token can interfere with each other through attention, weakening the bottleneck. Disabling their direct interaction restores performance gains (Fig. 2b).

**Contrastive self-distillation** is employed to directly refine the bottleneck feature self-conditioning operates on. A key insight is that diffusion models already have a high-quality "teacher" model through exponential moving averaging (EMA) of weights, a standard practice

| EMA rate | FID↓ | Acc.↑ |
|---|---|---|
| 0.999 | 9.17 | 60.06 |
| 0.9993 | 8.99 | 60.18 |
| **0.9999** | **8.81** | **60.47** |

to improve generation quality (Karras et al., 2024). Intriguingly, our preliminary study reveals that the EMA decay rate optimal for generation is also optimal for representation. This alignment suggests that EMA model, originally maintained solely for generation, serves as a powerful and readily available teacher for discriminative learning, obviating the need for a separate momentum encoder.

Leveraging this, we adapt MoCo v3 (Chen et al., 2021) for self-distillation. For an input view, features from the self-conditioning layer (*i.e.*, pooled or `[CLS]`) are passed through a time-dependent MLP projection head to handle the varying noise levels. These are trained to align with target features from EMA model, which are extracted from the same layer using an augmented view, but at a fixed timestep, the one used for probing. The final objective is a weighted sum of diffusion loss and contrastive loss: $\mathcal{L} = \mathcal{L}_{\text{diff}} + \gamma \mathcal{L}_{\text{MoCo}}$. Detailed formulations are provided in Appx. A.

**Summary.** Our full framework effectively unifies denoising autoencoding and instance discrimination within diffusion models based on an architectural enhancement. Studies with related concepts, such as SODA (Hudson et al., 2024) and REPA (Yu et al., 2025), are discussed in Appx. D.

Table 2: **A comprehensive comparison of generative and discriminative performance. Baseline:** Our standardized re-implementation across models and backbones to isolate design factors. **+ Self-conditioning:** Our method leverages discriminative features within the denoising network to guide generation by itself. As a generalizable enhancement, it improves one or both metrics in all settings, with the majority showing dual gains, especially evident on the more diverse CIFAR-100. Colored values indicate gains (or degradation). Best results for each backbone are in bold.

| Unconditional CIFAR-10 Generation & Linear Probing | | | | | | Unconditional CIFAR-100 Generation & Linear Probing | | | | | |
|---|---|---|---|---|---|---|---|---|---|---|---|
| Model | Backbone | Baseline | | + Self-conditioning | | Model | Backbone | Baseline | | + Self-conditioning | |
| | | FID↓ | Acc.↑ | FID↓ | Acc.↑ | | | FID↓ | Acc.↑ | FID↓ | Acc.↑ |
| DDPM | ddpm | 3.60 | 90.34 | 3.67 (+0.07) | 90.94 (+0.60) | DDPM | ddpm | 5.97 | 62.06 | **5.77** (-0.20) | 63.55 (+1.49) |
| EDM | ddpm | 3.39 | **91.41** | **3.30** (-0.09) | 91.07 (-0.34) | EDM | ddpm | 6.21 | 63.68 | 6.01 (-0.20) | **65.35** (+1.67) |
| RF | ddpm | 3.89 | 90.67 | 3.67 (-0.22) | 90.91 (+0.24) | RF | ddpm | 6.49 | 60.84 | 6.25 (-0.24) | 62.83 (+1.99) |
| DDPM | ddpmpp | 2.98 | 94.02 | 2.75 (-0.23) | 94.34 (+0.32) | DDPM | ddpmpp | 4.43 | 69.35 | 4.01 (-0.42) | 71.11 (+1.76) |
| EDM | ddpmpp | 2.23 | 94.83 | **2.18** (-0.05) | **94.85** (+0.02) | EDM | ddpmpp | 3.46 | 71.09 | **3.36** (-0.10) | **71.61** (+0.52) |
| RF | ddpmpp | 2.54 | 93.97 | 2.42 (-0.12) | 93.72 (-0.25) | RF | ddpmpp | 4.07 | 67.46 | 4.29 (+0.22) | 68.49 (+1.03) |
| DDPM | UViT-S | 4.48 | 93.67 | **4.13** (-0.35) | **94.30** (+0.63) | DDPM | UViT-S | 7.35 | 70.66 | **7.14** (-0.21) | **72.04** (+1.38) |

## 5 EXPERIMENTS

We present a series of experiments evaluating the efficacy, generalizability, scalability and the synergistic behavior of our method. In particular, we address the following research questions:

- How effective is our method in concurrently improving dual metrics on diverse baselines? (Tab. 2)
- How does each component interact and contribute to the overall improvements? (Tab. 3, Fig. 3)
- Does self-conditioning remain effective, scalable, and transferable with DiT trained on ImageNet? (Fig. 4, Tab. 4, 5) How does it work? (Fig. 5, 6, 7)

**Implementation details.** Our pixel-space experiments build upon DDAE (Xiang et al., 2023), after standardizing hyper-parameters in linear probing (*e.g.*, sufficient epochs and refined learning rates) to ensure fair comparison across models. Consequently, Tab. 2 represents our new controlled setup, so we do not report DDAE's original results. For latent-space models, we follow the state-of-the-art Lightning-DiT (Yao et al., 2025). FID (Heusel et al., 2017) and IS (Salimans et al., 2016) are used to measure sample quality. Linear probing is used to measure feature quality, where the backbone is frozen without any fine-tuning. Details on training, evaluation, layer/timestep hyper-parameters and other SSL methods in comparison are specified in Appx. B, C, and qualitative results are in Appx. E.

### 5.1 MAIN PROPERTIES

**Observations from baseline models.** Tab. 2 shows the performance of our re-implemented, unconditional baselines, encompassing combinations[2] of models and backbones described in Sec. 3. Regarding sample quality, EDM generally achieves best FID, consistently outperforming RF, another continuous-time model, suggesting that designing a suitable noise schedule is crucial. Meanwhile, UViT still lags behind UNet in pixel-space generation. For classification, while larger models tend to yield higher results, the model formulation is also crucial. Notably, EDM consistently delivers the highest accuracy, even when it does not achieve the lowest FID (*e.g.*, EDM-`ddpm` on CIFAR-100), suggesting that generative and discriminative qualities do not naturally align in baseline models.

**Self-conditioning generally improves both metrics.** Tab. 2 shows a dual-improvement trend: it not only improves FID but also boosts feature quality, particularly on the more diverse CIFAR-100 where accuracy increases by up to 1.99%, a substantial gain in the SSL context. Remarkably, this is achieved with negligible overhead: a mere 0.5M parameters for UNet and one more token for UViT.

We now extend our analysis to assess how self-conditioning interacts with discriminative techniques, namely data augmentations (Aug) and contrastive learning (CL), with results presented in Tab. 3.

**Aug alone provides limited benefits.** Using Aug provides mixed results: it slightly improves FID on CIFAR-10 but can degrade it notably on CIFAR-100. This suggests the geometric pipeline, tuned by EDM for CIFAR-10, may not generalize well, highlighting a key challenge for diffusion-based unified learning. Crucially, the fact that FID can worsen even when feature quality improves (*e.g.*, 7.35 to 7.96) indicates a disconnect: enhanced features are not effectively leveraged for generation.

---

[2]We find UViT to be unstable with EDM and RF, so these two are excluded.

Table 3: **Component-wise analysis and comparison with other SSL methods.** Self-conditioning, when using in conjunction with Aug and/or CL, provides additional improvements to surpass other self-supervised or diffusion-based models. Best results for each backbone are in bold.

| Model | Backbone | Aug | CL | CIFAR-10 | | CIFAR-100 | |
|---|---|---|---|---|---|---|---|
| | | | | FID↓ | Acc.↑ | FID↓ | Acc.↑ |
| EDM | ddpm | | | 3.39 | 91.41 | 6.21 | 63.68 |
| | | ✓ | | 3.32 | 92.55 | 5.39 | 66.28 |
| | | ✓ | ✓ | 3.42 | 92.61 | 5.42 | 67.63 |
| | + 0.46M | | | 3.30 | 91.07 | 6.01 | 65.35 |
| | (1.3%) | ✓ | | **3.08** | 92.41 | **5.13** | 68.65 |
| | | ✓ | ✓ | 3.14 | **92.97** | 5.25 | **69.50** |
| EDM | ddpmpp | | | 2.23 | 94.83 | 3.46 | 71.09 |
| | | ✓ | | 2.19 | 95.24 | 3.61 | 71.14 |
| | + 0.46M | | | 2.18 | 94.85 | 3.36 | 71.61 |
| | (0.8%) | ✓ | | 2.17 | 95.33 | 3.38 | **73.29** |
| | | ✓ | ✓ | **2.14** | **95.35** | **3.35** | 72.88 |
| DDPM | UViT-S | | | 4.48 | 93.67 | 7.35 | 70.66 |
| | | ✓ | | 4.00 | 95.14 | 7.96 | 72.85 |
| | + 1 token | | | 4.13 | 94.30 | 7.14 | 72.04 |
| | (0.4%) | ✓ | | **4.00** | **95.34** | 7.05 | 73.58 |
| | | ✓ | ✓ | 4.35 | 95.28 | **6.88** | **74.48** |
| Contrastive | resnet18 | ✓ | ✓ | 89.17–93.10 | | 64.88–70.90 | |
| | resnet50 | ✓ | ✓ | 90.88–93.89 | | 66.15–72.51 | |
| MIM-based | ViT-B | crop | | 61.70–70.20 | | — | |
| MDM | unet | | | 94.80 | | — | |
| SODA[#] | res18+unet | ✓ | | 80.00 | | 54.90 | |

[#]Official code unavailable; we build a simplified version based on core principles.

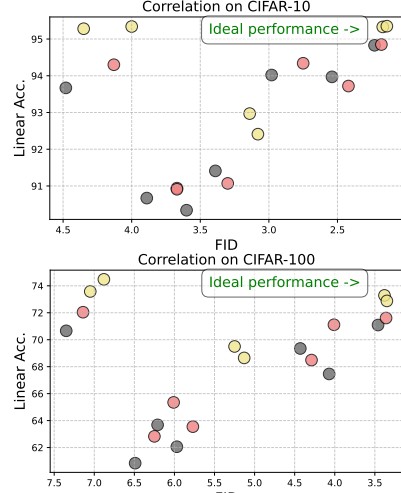

Figure 3: **Correlation between generation and discrimination.** Original diffusion baselines (in gray) only show relatively weak linear correlation. Our self-conditioning, aug and contrastive enhancements make it more significant by simultaneously and gradually boosting both metrics towards the upper-right ideal region.

**Self-conditioning creates the necessary architectural bridge.** When applied jointly with Aug, it ensures that improved representations are effectively *translated into generative benefits*: enhancing features alone may not be sufficient if the model cannot leverage it. It not only counteracts the FID degradation on CIFAR-100 (*e.g.*, from 7.96 back to 7.05, and 3.61 to 3.38), but also amplifies gains in both domains where Aug alone is marginally helpful (*e.g.*, boosting 71.09-71.14% to 73.29%).

**CL and diffusion are complementary.** While adding CL can sometimes cause a slight trade-off with FID, it is mitigated by self-conditioning, and it suggests the information generated by CL may differ from that needed in diffusion. This can actually be beneficial to building stronger representations: our fully combined method consistently achieves high accuracy (the highest in most cases).

**Putting all together.** Fig. 3 visualizes the evolving relationship between generation and discrimination. In baseline models, the correlation (discussed in Xiang et al. (2023); Yu et al. (2025)) is relatively weak, indicating that better generative models do not necessarily yield better representations (in line with Chen et al. (2025)). In contrast, our method establishes a much clearer positive correlation, demonstrating its ability to structurally align the two domains. This effect is particularly pronounced for UViT. Although the accuracy is initially outperformed by UNet, our enhancements unlock its potential, enabling it to surpass ddpmpp with less compute and achieve highest results in Tab. 3. This suggests that, despite lagging behind UNet in pixel-space generation, UViT possesses unique potential to learn powerful representations, which we successfully *unlock*, even in this low-data regime where ViTs typically struggle (Dosovitskiy et al., 2021).

## 5.2 PERFORMANCE AND SCALABILITY ON IMAGENET

Following standard practices (Chen et al., 2025; Yu et al., 2025), we conduct experiments on the more challenging ImageNet-1k ($256^2$) with latent-space DiT. Our models show consistent performance gains across diverse settings, including class-conditional and unconditional generation, with classifier-free guidance (CFG) (Ho & Salimans, 2021), scaling from DiT-B to DiT-L, and scaling training epochs. Fig. 4 reveals a comprehensive picture: our models consistently outperform their respective baselines in FID, IS and accuracy throughout the training. Most importantly, the performance gap does not narrow; in fact, the *accuracy gain widens as training progresses*, highlighting a desirable **scaling property** similar to other generative methods (He et al., 2022; Chen et al., 2025).

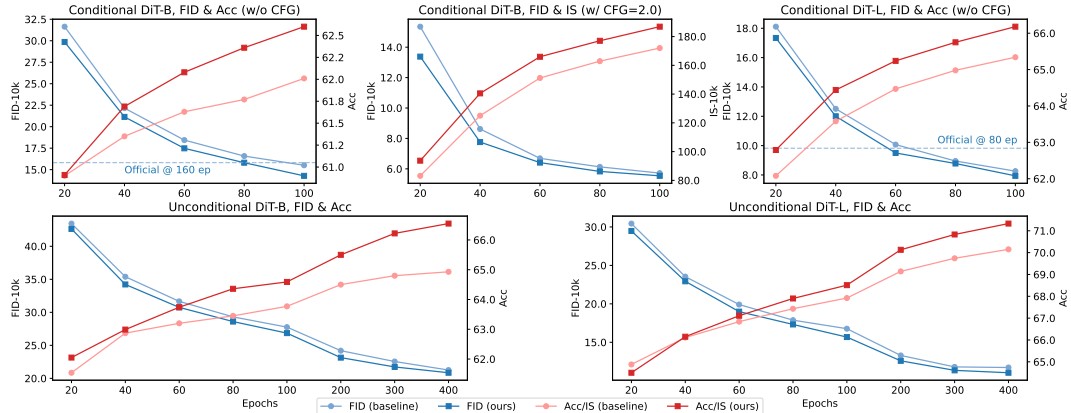

Figure 4: **Detailed performance evolution.** Our method maintains a consistent lead over the corresponding baselines, with continuously widening accuracy gaps. FID/IS-10k is used for efficiency.

**Comparison with SSL methods.** While our models do not yet match leading CL models (Tab. 4), they outperform other generative learners such as l-DAE (Chen et al., 2025) and MIM (Shin et al., 2024), and DINO without strong augmentations (Hudson et al., 2024). We believe a key factor limiting our recognition accuracy is *the absence of augmentation in latent-space training* (at least random crop), a non-trivial challenge for latent diffusion that remains an open direction.

**Comparison with advanced diffusion systems.** Our baseline on Lightning-DiT (Yao et al., 2025) is already highly-optimized and, with REPA (Yu et al., 2025), represents the state-of-the-art with extremely fast convergence. Against this strong baseline, it is compelling that our minimal tweak yields further gains. Note that while REPA achieves high accuracy via direct DINOv2 alignment, this gain does not fully translate into a superior FID, a disconnect similar to Sec. 5.1. This suggests that *externally boosting feature quality alone is insufficient, if the internal decoder remains architecturally decoupled from discriminative gains.* We identify this missing link and explore an alternative *ab initio* way, aiming to structurally bridge it.

Table 4: **Comparison on ImageNet-1k.** Diffusion models only use horizontal flip; other methods use stronger augmentations.

| Model | Backbone | Aug | CL | ImageNet-1k | | |
|---|---|---|---|---|---|---|
| | | | | Ep. | FID[†]↓ | Acc.↑ |
| Flattened, clean VAE latent | | | | | | 40.04 |
| RF | DiT-B | | | 400 | 18.86 | 64.93 |
| | +1.38M | | | 400 | 17.94 | 66.55 |
| RF | DiT-L | | | 400 | 8.73 | 70.15 |
| | +2.36M | | | 400 | 8.07 | 71.33 |
| RF (cond) | DiT-B | | | 100 | 2.87–12.70 | 62.01 |
| | +1.38M | | | 100 | 2.67–11.89 | 62.60 |
| RF (cond) | DiT-L | | | 100 | 2.09–5.53 | 65.34 |
| | +2.36M | | | 100 | **2.07–5.34** | 66.18 |
| RF (cond) | DiT-XL | | | 40 | 7.69 | 64.87 |
| | +2.95M | | | 40 | 7.34 | 65.36 |
| REPA (cond) | DiT-B | | ✓* | 80 | 24.40 | 61.20 |
| | DiT-L | | ✓* | 80 | 10.00 | 69.40 |
| | DiT-XL | | ✓* | 40 | 11.10 | 67.30 |
| l-DAE | DiT-L | crop | | 400 | 11.60 | 57.50 |
| | DiT-L | crop | | 400 | — | 65.00 |
| | DiT-Bx2 | crop | | 400 | — | 60.30 |
| CL | ViT-B | ✓ | ✓ | 300 | | 71.60–**78.20** |
| CL[#] | ViT-B | crop | ✓ | 400 | | 61.10–65.30 |
| MIM | ViT-B | crop | | 400 | | 61.40–62.90 |
| | ViT-L | crop | | 200 | | 62.20–65.80 |

[†]FID-50k; denoted as (w/ CFG –) w/o CFG for conditional models.
*Aligning representations to an external DINOv2-B encoder.
[#]DINO with only flip+crop (crop – multi-crop), reported by SODA.

**Transfer learning on semantic segmentation.** Beyond a stronger global representation, our approach also enhances local features transferable to dense tasks. We validate this on $256^2$ versions of ADE20K (Zhou et al., 2017) and VOC2012 (Everingham et al., 2010), following dense linear probing in iBOT (Zhou et al., 2022) and DINOv2 (Oquab et al., 2024). A linear head is trained to predict logits from patch tokens ($16^2$), which are upsampled to obtain segmentation map. Tab. 5 shows that our method consistently outperforms baselines: on ADE20K, it quickly converges to the performance upper bound in this challenging setting; on VOC, it outpaces the baseline with a continuously widening mIoU gain (0.73, 0.88 to 1.27, similar to Fig. 4), while narrowing the gap to DINO with higher resolution and low patch size.

Table 5: **Linear segmentation with frozen features.**

| DiT-B | ADE20K | VOC2012 |
|---|---|---|
| Pre-train ep. | mIoU↑ | mIoU↑ |
| 100 | 29.75 | 62.53 |
| + Self-cond. | 30.40 | 63.26 |
| 200 | 30.07 | 63.45 |
| + Self-cond. | 30.72 | 64.33 |
| 400 | 30.48 | 64.34 |
| + Self-cond. | **30.75** | **65.61** |
| ViT-B/8 (DINO, $512^2$) | 31.80 | 66.40 |

**How does self-conditioning work?** Fig. 5 and Fig. 6 show how it operates. First, we focus on how it reshapes the layer-wise linear separability: while the baseline exhibits a shift of discriminative power towards middle layers as training progresses (gradual reallocation of encoder-decoder capacity), the

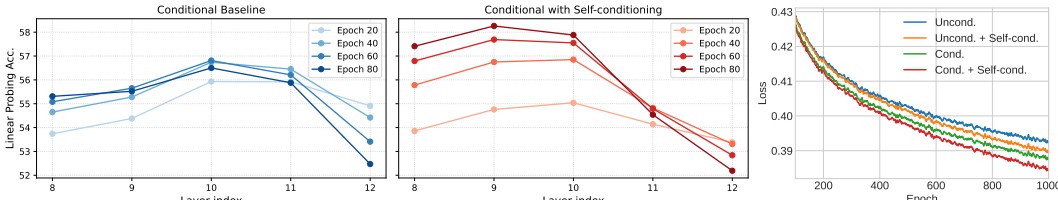

Figure 5: **Self-conditioning reshapes feature distribution.** Compared to dispersed baseline, it accelerates the convergence of discriminative features to middle layers, forming a pronounced concentration. Probes are trained for fewer epochs to better reveal relative differences.

Figure 6: **Self-conditioning facilitates the optimization** and reduces the loss for both un- and class-cond. training.

shift is slow and the variation is modest, in line with the distributed representation theory in Sec. 1. In contrast, our method accelerates this convergence and creates a pronounced hierarchy. This sharper focus suggests a more effective bottleneck and a more decoupled decoder. Intriguingly, while we inject features after the 10th layer in this case, peak accuracy emerges earlier, indicating that the most discriminative features undergo further transformation to become optimal as guidance signals. See Fig. 7 below for deeper mechanistic analysis based on feature similarity, which directly demonstrates the functional decoupling and multiple phases caused by our bottleneck design. Additionally, the observed reduction in denoising loss suggests that effective semantic guidance is indeed taking place.

### 5.3 VISUALIZING BOTTLENECK FORMATION

To validate our hypothesis that self-conditioning reshapes the model's semantic hierarchy and forms a more explicit semantic bottleneck, we analyze the feature similarity using Centered Kernel Alignment (CKA) (Kornblith et al., 2019). We use DiT-L with more layers (24) for better visualization.

**Intra-model similarity indicates decoupling.** In the baseline model (Fig. 7), the similarity matrix exhibits a smooth and gradual decay as depth increases. While a faint distinctness appears at Layer 12, the transition remains smooth afterwards, and there is no clear structural boundary between two adjacent layers that would separate a semantic encoding phase from a generative decoding phase. This continuous spectrum suggests that diffusion models perform generation through a gradual evolution of features, with encoding and decoding roles entangled throughout most of the layers.

In contrast, our model (self-conditioned after Layer 17) shows a pronounced **phase transition** at the injection point. First, Layers 1-17 maintain high similarity, indicating a coherent process of accumulating global context to form the guidance signal. Then, a sharp decrease in similarity appears between these layers and the subsequent layers (18-24). We attribute this phenomenon to the altered dependency structure: since the decoding layers are explicitly modulated by the concentrated semantics from Layer 17, the hidden states are no longer required to preserve information purely

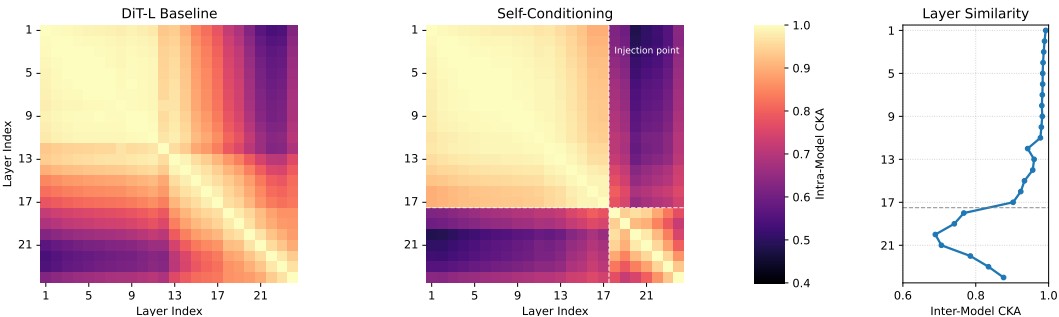

Figure 7: **CKA analysis of representation structure** reveals that self-conditioning induces a structural phase transition absent in the baseline: a semantic encoding phase (Layers 1-17) followed by a specialized, decoupled generative decoding phase (Layers 18-24) modulated by the bottleneck. The high similarity within the encoder's late layers (14-17) suggests an implicit adapter, transforming the most discriminative features (peaking earlier) into optimal guidance signals right before injection.

Table 6: **Ablation studies** on CIFAR-10/100 for 1200 epochs. Our design choices are in gray.

(a) Self-conditioning in UNet.

| Models, approaches | FID↓ | Acc.↑ |
|---|---|---|
| DDPM, baseline | 3.02 | 94.02 |
| DDPM, addition | 2.80 | 94.18 |
| DDPM, adaptive | **2.76** | **94.34** |
| EDM, baseline | 2.25 | 94.83 |
| EDM, addition | 2.43 | 94.85 |
| EDM, adaptive | **2.21** | **94.85** |
| RF, baseline | 2.57 | **93.97** |
| RF, addition | 2.69 | 93.60 |
| RF, adaptive | **2.42** | 93.72 |

(b) MLP projection head.

| MLP head | FID↓ | Acc.↑ |
|---|---|---|
| Original MoCo v3 | 5.87 | 69.01 |
| Time-dependent | **5.88** | **69.50** |

(c) Distillation timestep.

| Target timestep | FID↓ | Acc.↑ |
|---|---|---|
| Minimal noise | 5.81 | 68.76 |
| Optimal for linear | **5.88** | **69.50** |

(d) Contrastive loss weight.

| $\gamma$ | FID↓ | Acc.↑ |
|---|---|---|
| 0.1 | 6.03 | 69.08 |
| 0.01 | **5.88** | **69.50** |
| 0.001 | 5.96 | 68.42 |

(e) Distillation target in UViT.

| Feature to contrast | FID↓ | Acc.↑ |
|---|---|---|
| Pooling of tokens | 8.51 | 73.15 |
| [CLS] | **7.37** | **74.48** |

through direct layer-to-layer inheritance. Consequently, representations in these final layers are free to diverge rapidly and specialize in the local denoising operations required near the output end.

**Misalignment between peak accuracy and injection point.** As noted in Sec. 5.2, the layer yielding the highest linear probing accuracy (Layer 13 in this DiT-L case) may precede our self-conditioning layer. The CKA map, however, reveals high similarity among features in Layers 14-17. This suggests that while discriminative power peaks earlier, the underlying information content remains relatively stable across this interval, which functions as an implicit **adapter**: these blocks transform the concentrated discriminative features into a format more optimal for generative guidance, preparing the representation for subsequent injection without fundamentally losing its semantic core.

**Inter-model divergence.** Finally, a layer-wise comparison between the baseline and our model confirms that their representations begin to significantly diverge immediately after the self-conditioning layer. This confirms that our mechanism actively intervenes in the output stage, reshaping the hidden states in specialized decoding layers to leverage the enhanced semantic guidance.

## 5.4 ABLATION STUDIES

Below we show ablation on design choices in self-conditioning and self-distillation. Tab. 6a: The adaptive addition of semantic features into the pathway is crucial, which consistently yields greater improvements, whereas a direct addition does not show much benefit. Tab. 6b: Using the original MoCo v3 projection head results in the same FID, but linear probing accuracy decreases by 0.5% compared to our time-dependent one. Tab. 6c: Using target features extracted with the optimal timestep in linear probing, leads in accuracy by 0.7% when compared to SD-DiT's minimal-noise design (Zhu et al., 2024). Tab. 6d: Either excessive and insufficient contrastive weight leads to degradation in both metrics, suggesting that CL can contribute positively to dual aspects when appropriately tuned. Tab. 6e: We investigate two potential distillation target in UViT: from global average pooling or the [CLS] token. The [CLS] approach benefits both metrics, indicating that self-conditioning based on this token works well because it aggregates global semantics naturally.

Finally, we validate our layer selection strategy with DiT-B. Among four candidates, the one yielding the lowest training loss after just 20 epochs also achieves the best for both FID and Acc after 100 epochs, indicating that early training loss is highly predictive of final relative performance, and this strategy is efficient and principled. It is also intriguing that

| Layer index | Loss↓ (20 ep) | FID-10k↓ (100 ep) | Acc.↑ (100 ep) |
|---|---|---|---|
| baseline | 0.4141 | 15.51 | 62.01 |
| 8 | 0.4135 | 14.79 | 62.47 |
| 9 | **0.4133** | **14.28** | **62.60** |
| 10 | 0.4136 | 14.52 | 62.39 |
| 11 | 0.4140 | 14.77 | 62.19 |

a single time-averaged loss correlates with both metrics, despite their distinct time-dependencies.

## 6 CONCLUSION

We show that established diffusion architectures can be enhanced by simply conditioning the decoding process on features learned by themselves. Beyond merely improving representation quality, our self-conditioning principle forges the architectural bridge necessary to translate discriminative gains into tangible generative improvements, a crucial but previously under-explored step. We hope this work contributes to the ongoing exploration of diffusion architectures and unified learning.

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

# A  DESCRIPTION OF OBJECTIVES

**DDPM** (Ho et al., 2020) operates over a large number of discrete timesteps ($T = 1000$), based on a variance preserving schedule, *i.e.*, $x_t = \alpha_t x_0 + \sqrt{1 - \alpha_t^2}\epsilon$, where $\alpha_t = \sqrt{\Pi_{i=1}^{t}(1 - \beta_i)}$ and $\beta_{1:T}$ is pre-defined with a linear schedule. The network learns to predict the noise $\epsilon_\theta$, and is trained using the $\epsilon$-prediction objective $\|\epsilon_\theta(x_t, t) - \epsilon\|_2^2$. Sampling is performed with Euler's method (also known as DDIM (Song et al., 2021a)), generating images in 100-250 discretization steps.

**EDM** (Karras et al., 2022) is a continuous-time model based on the variance exploding parameterization, *i.e.*, $x_t = x_0 + \sigma_t \epsilon$, where $\sigma_t$ spans a continuous range of $[0.002, 80]$. The network predicts the denoised image $D_\theta$, and is trained with the $x_0$-prediction objective $\|D_\theta(x_t, t) - x_0\|_2^2$. For sampling, the 2$^{\text{nd}}$ order Heun solver can efficiently generate images in 18-40 discretization steps, though the actual neural function evaluations (NFE) are doubled.

**RF** (Rectified Flow) (Liu et al., 2023; Lipman et al., 2023) defines the forward process as a linear interpolation between data and noise, *i.e.*, $x_t = (1 - t)x_0 + t\epsilon$, with $t$ sampled from $[0, 1]$. The network directly estimates the velocity and is trained using a $v$-prediction objective $\|v_\theta(x_t, t) - (\epsilon - x_0)\|_2^2$. Euler's or an adaptive RK45 ODE solver is used for sampling within 100-250 steps.

**MoCo v3** (Chen et al., 2021) is a contrastive self-distillation framework that learns representations by matching positive pairs. It uses an online model to compute the representation $q_1$ of an image, and a momentum-updated teacher for representation $k_2$ of another augmented view. Both online model and teacher consist of a backbone and a MLP projection head (Chen et al., 2020b), and the online encoder has an extra MLP prediction head (Grill et al., 2020). MoCo v3 employs a symmetrized contrastive loss $\mathcal{L}_{\text{MoCo}} = \mathcal{L}_{\text{InfoNCE}}(q_1, k_2) + \mathcal{L}_{\text{InfoNCE}}(q_2, k_1)$ to train the model. In our framework, the projection head is made to be time-dependent, in order to prevent conflicts across different noise levels. This is achieved by adding a time embedding to the input feature before the first linear layer.

# B  IMPLEMENTATION DETAILS

**Pixel-space diffusion models.** Our implementations are primarily based on official DDAE[3] and UViT[4] codebases, covering DDPM, EDM, `ddpm`, `ddpmpp`, and UViT. We use the uncertainty loss weighting from EDM2 (Karras et al., 2024) for EDM with augmentations to accelerate convergence. We implement RF following Liu et al. (2023); Lipman et al. (2023) and Yao et al. (2025). Our `ddpm` and UViT are equivalent to official ones, while `ddpmpp` is a simplified variant of DDPM++ in Song et al. (2021b) that omits certain details like skip connection rescaling, as we did not find beneficial.

**Latent-space diffusion models.** Our implementations of DiT-B/L strictly follow the official repository of Lightning-DiT[5], including its pre-trained VA-VAE. We also fix the "3-channel CFG" bug originated in DiT's code to enhance sample quality. More details are provided in Tab. 7 and Tab. 8.

Table 7: **Details of diffusion model formulations.**

| Pixel-space DDPM | |
|---|---|
| $T$ | 1000 |
| training noise schedule | linear beta schedule $[10^{-4}, 0.02]$ |
| training loss weighting | none |
| dropout rate | 0.1 |
| EMA decay rate | 0.9999 |
| ODE sampler | Euler (DDIM) |
| sampling steps | 100 |

| Pixel-space RF | |
|---|---|
| training noise schedule | $t \in [0, 1]$ |
| training loss weighting | none |
| dropout rate | 0.1 |
| EMA decay rate | 0.9999 |
| ODE sampler | adaptive RK45 |
| sampling steps | 140-160 |

| Pixel-space EDM | |
|---|---|
| training noise schedule | $P_{mean} = -1.2, P_{std} = 1.2$ |
| training loss weighting | uncertainty (when using Aug) |
| dropout rate | 0.13 |
| EMA decay rate | 0.9993 |
| ODE sampler | 2$^{\text{nd}}$ order Heun |
| sampling noise schedule | $[\sigma_{min}, \sigma_{max}] = [0.002, 80], \rho = 7$ |
| sampling steps | 18 (35 NFEs) |

| Latent-space RF | |
|---|---|
| training noise schedule | $t \in [0, 1]$ , lognorm sampling |
| EMA decay rate | 0.9999 |
| ODE sampler | Euler |
| sampling steps | 250 |
| CFG scale | 2.0 (DiT-B), 1.4 (DiT-L) |
| CFG interval | $[0, 0.89]$ |
| timestep shift | 3.3 |

---

[3]https://github.com/FutureXiang/ddae

[4]https://github.com/baofff/U-ViT

[5]https://github.com/hustvl/LightningDiT

**Pre-training.** For pixel-space experiments on CIFAR, all models are trained for 2000 epochs. Checkpoints are evaluated every 200 epochs to find the best FID, typically achieved between 1400-2000 epochs. Training setups for DDPM and RF are identical, while EDM uses slightly different warmup, dropout, and EMA rates following its original paper. For UViT, we find that excluding bias and positional embeddings from weight decay, while applying it to `[CLS]`, can further improve results. For latent-space experiments on ImageNet, DiT models are trained for 400 epochs (unconditional) or 100 epochs (class-conditional). Horizontal flip is the only data augmentation used, so we pre-cache the two versions of latent codes efficiently.

Table 8: **Details of backbone architectures.**

| UNet-based architecture | ddpm | ddpmpp | |
|---|---|---|---|
| base channels (multiplier) | 128 | 128 | |
| channels per resolution | 1-2-2-2 | 2-2-2 | |
| blocks per resolution | 2 | 4 | |
| attention resolutions | {16} | {16} | |
| attention heads | 1 | 1 | |
| BigGAN up/down blocks | no | yes | |
| (param count) | 35.7M | 56.5M | |
| *ViT-based architecture* | **UViT-S** | **DiT-B** | **DiT-L** |
| patch size | 2 | 1 | 1 |
| hidden size | 512 | 768 | 1024 |
| ViT layers | 13 | 12 | 24 |
| attention heads | 8 | 12 | 16 |
| (param count) | 44.3M | 129.8M | 457.1M |
| *Latent-space compressor* | **VA-VAE** | | |
| output latent size | 16x16 (256/16) | | |
| output dimension | 32 | | |

**Feature extraction protocols.** In our work, feature extraction involves several distinct layer selection processes depending on the context. We distinguish between choices made for model training (which must be principled to avoid data leakage from test/validation sets) and those made for evaluation (which follow common practices to measure a model's peak performance):

- **Self-conditioning layer** (training-time): For DiT, this layer is chosen *before full training*. We use the method described in Sec. 4.1: selecting the one with lowest training loss in short profiling runs. For UNet models, since faithfully applying this to all 7 variants is complicated, we adopt a transfer protocol: setting this layer based on the baseline's optimal probing layer. This simplification is reasonable, as our DiT experiments show that these two layers are typically close. Furthermore, our goal with the extensive UNet experiments is to demonstrate a general observation across models, rather than to tune each individual variant for its peak performance.
- **Linear probing** (evaluation-time): Following Baranchuk et al. (2022); Xiang et al. (2023), we determine the timestep and layer by grid searching on a validation set. This process is performed independently for every model (baselines and ours) to identify its highest possible classification performance. Note that this layer in our model may change from its baseline due to self-conditioning.
- **Semantic segmentation** (evaluation-time): Following iBOT (Zhou et al., 2022), to leverage multi-level features for higher performance, we extract patch tokens from two fixed intermediate layers (5th and 7th, for both baselines and ours) and concatenate them. The optimal timesteps obtained by searching are 0.25 (ADE20K) and 0.45 (VOC), slightly noisier than the one in classification.

The specific timesteps and layer indices are summarized in Tab. 9. The optimal timestep found for linear probing is also used for extracting the target feature in contrastive self-distillation.

Table 9: **Time and layer for feature extraction.** "out" denotes output layers in UNet and UViT.

| Model | Backbone | *On CIFAR-10* | | *On CIFAR-100* | |
|---|---|---|---|---|---|
| | | **Time** | **Layer** | **Time** | **Layer** |
| DDPM | ddpm | $t = 11$ | out_6/12 | $t = 11$ | out_6/12 |
| EDM | ddpm | $\sigma_t = 0.06$* | out_7/12 | $\sigma_t = 0.06$* | out_6/12 |
| RF | ddpm | $t = 0.06$ | out_7/12 | $t = 0.06$ | out_7/12 |
| DDPM | ddpmpp | $t = 11$ | out_7/15 | $t = 11$ | out_8/15 |
| EDM | ddpmpp | $\sigma_t = 0.06$* | out_9/15 | $\sigma_t = 0.06$* | out_8/15 |
| RF | ddpmpp | $t = 0.06$ | out_8/15 | $t = 0.06$ | out_8/15 |
| DDPM | UViT-S | $t = 11$ | out_2/6 | $t = 11$ | out_2/6 |

*Corresponds to $t = 4$ in the 18-step EDM sampling schedule.

| Model | Backbone | *On ImageNet-1k* | |
|---|---|---|---|
| | | **Time** | **Layer*** |
| RF | DiT-B | $t = 0.25$ | 9-9-7/12 |
| | DiT-L | $t = 0.25$ | 16-17-13/24 |
| (cond) | DiT-B | $t = 0.25$ | 10-9-6/12 |
| | DiT-L | $t = 0.25$ | 8-9-11/24 |

*Denoted as "A-B-C/Total": **A**: Optimal probing layer for baseline. **B**: Chosen layer for self-conditioning. **C**: New optimal probing layer for self-conditioned model.

**Linear and dense probing.** We simplify the settings from DDAE by using identical training epochs and learning rates for all UNet models. We also found that random cropping is not helpful enough in linear probing, so we use only horizontal flipping as augmentation. On CIFAR, the highest accuracy across checkpoints at 800, 1000 and 1200 pre-training epochs is reported. On ImageNet, the performance does not saturate till the end of 100/400-epoch pre-training. A linear classifier (with parameter-free BatchNorm following Chen et al. (2025)) is trained on top of the frozen, global average pooled features. For dense probing (semantic segmentation), no data augmentation is used, and a linear head (with LayerNorm) is trained on top of the frozen patch token features, followed by 16x upsampling to produce the final map. Detailed configurations are shown in Tab. 10.

Table 10: **Details of pre-training and linear evaluations.**

*On CIFAR-10, CIFAR-100*

| | Pre-training | Linear probing |
|---|---|---|
| GPUs | 4 * NVIDIA 3080Ti GPU | |
| batch size | 512 (ddpm) or 256 (ddpmpp, uvit) | |
| epochs | 2000 | 15 |
| warmup epochs | 200 (EDM) / 13 (others) | — |
| optimizer | Adam (unet) AdamW (uvit) | Adam |
| learning rate | 4e-4 | 4e-3 |
| lr schedule | constant | cosine |
| augmentations | flip (or non-leaky) | flip |

*On ImageNet-1k*

| | Pre-training | Linear probing | Dense probing |
|---|---|---|---|
| GPUs | 6 * NVIDIA 4090 GPU | | |
| batch size | 1024 | 1440 | 128 |
| epochs | 100-400 | 30 | 30 |
| optimizer | AdamW in LightningDiT | Adam | Adam |
| learning rate | 2e-4 | 2e-3(B)/1e-3(L) | 3e-3 |
| lr schedule | constant | cosine | cosine |
| augmentations | flip | flip | none |

## C CLARIFICATIONS ON BASELINES

We prioritize *fair, controlled comparisons with corresponding baselines* over chasing absolute state-of-the-art numbers. Here we clarify our benchmarking protocols and contextualize the results.

**Sources of reference SSL baselines.** We report their performance as a range covering representative works, with each result retrieved directly from the original papers or established benchmarks. Specifically: On CIFAR (Tab. 3), contrastive results cover numbers from SimCLR (Chen et al., 2020b), BYOL (Grill et al., 2020), SwAV (Caron et al., 2020), DINO (Caron et al., 2021), MoCo v3 (Chen et al., 2021), SimSiam (Chen & He, 2021), Barlow (Zbontar et al., 2021), VICReg (Bardes et al., 2022), and benchmarks in Da Costa et al. (2022); Bandara et al. (2023). MIM results include MAE (He et al., 2022) and U-MAE (Zhang et al., 2022). MDM is from Pan et al. (2023). For SODA (Hudson et al., 2024), as the official code is unavailable, we implemented a version adhering to its core principles (bottleneck within a ResNet encoder and a UNet decoder). Note that SODA's design utilizes strong augmentations or distortions, and we found it harmful for unconditional generation on CIFAR; we report the best result we achieved. On ImageNet (Tab. 4), contrastive results are similarly collected from SimCLR, BYOL, SwAV, DINO, MoCo v3, SimSiam, Barlow, VICReg, and CorInfoMax (Ozsoy et al., 2022). MIM includes MAE, U-MAE, and SGMAE (Shin et al., 2024).

**Worse EDM FID (2.19 *vs.* 1.97).** Our re-implemented EDM+Aug baseline on CIFAR-10 achieves an FID of 2.19 (Tab. 3), compared to the official 1.97. This gap mainly results from our standardized training setup: while the original EDM trained for 4000 epochs, we unified all 7 variants (DDPM, EDM, RF $\times$ backbones) to DDPM's 2000 epochs (Ho et al., 2020) for fairness. Independent study (Adaloglou et al., 2025) also confirms that EDM yields FID >2.2 after 2000 epochs (close to ours, and their CIFAR-100 results also align with ours), and arrives at only 2.07 after 4000 epochs. Discussions in EDM's official repository also confirm this reproducibility difficulty[6]. Our numbers are thus reasonable, and our key contribution lies in the consistent improvement over rigorous baselines.

**FID on ImageNet.** Our reported FIDs might appear higher than absolute SOTA values, since we use DiT-B/L with 100/400 epochs to efficiently examine effectiveness. Rather than training DiT-XL for 800 epochs for an absolute SOTA, we focus on the consistent gains across model sizes and training durations. As shown in Fig. 4 and Tab. 5, our method exhibits a desirable trend where *discriminative gains widen* as training durations scale up, validating its potential for larger settings.

While we cannot claim SOTA-level performance without a full-scale run due to compute constraints, current results strongly suggest that mechanisms like self-conditioning would be a valuable component for diffusion models. Verifying this at larger scales is an important direction for future work.

## D METHODOLOGY COMPARISON

Our full DDAE++ framework combines two core principles of SSL: **cross-view alignment** from discriminative contrastive learning and **intra-view reconstruction** inherent in denoising models. For generative modeling, it establishes a **self-contained semantic-conditional** mechanism that uses intermediate features to guide the generation process, with the feature encoder absorbed into the first few layers in network. In this section, we provide a detailed comparison with closely related works.

---

[6]https://github.com/NVlabs/edm/issues/16

**Relation to SODA.** SODA (Hudson et al., 2024) learns representations through cross-view reconstruction. Similar to self-conditioning, it also employs a feature modulation mechanism to impose a tighter bottleneck between the encoder and decoder, thereby learning compact, linearly-separable features. However, SODA focuses on image-conditional tasks like novel view synthesis, so it uses a disentangled encoder separate from diffusion decoder. Additionally, SODA's features are learned through pure generative pre-training, without investigating the influence of contrastive methods.

A direct comparison with SODA is not so appropriate, since SODA is optimized for representations and *cannot* function as a regular diffusion model in standard unconditional (or class-conditional) settings. Moreover, SODA may not achieve superior FID and acc. *simultaneously* with a same model, as it employs different augmentations for classification (stronger) and reconstruction (weaker).

**Relation to SD-DiT.** SD-DiT (Zhu et al., 2024) aims to accelerates DiT training through self-distillation. It aligns features extracted from the visible patches of an image with those extracted from the entire image by an EMA teacher. While this joint optimization of generative and discriminative objectives is similar to our approach, it does not focus on enhancing representation quality. Furthermore, its key design, setting the distillation target to the minimal noise scale, differs from ours that using the linear probing timestep, proven more effective in ablation.

**Relation to REPA.** REPA (Yu et al., 2025) is a recent work that accelerates DiT training through representation enhancement. It shows that aligning the features in diffusion models with vision foundation models can significantly ease training. This "representation for better generation" idea is similar to ours, but it uses an external DINOv2 as the teacher, rather than leveraging diffusion models themselves. Additionally, REPA employs MLP heads and similarity functions to align features, similar to our MoCo v3-based method. However, it does not introduce two views for contrastive learning, as it is essentially a knowledge distillation process rather than a self-supervised one.

## E    QUALITATIVE RESULTS

To visualize the generation quality, we present randomly generated samples in Fig. 8 (CIFAR-100) and Fig. 9 (ImageNet-1k) using the same set of initial noise inputs.

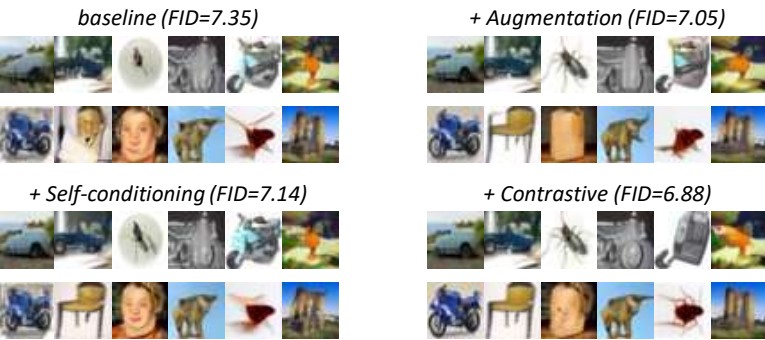

*baseline (FID=7.35)*          *+ Augmentation (FID=7.05)*

*+ Self-conditioning (FID=7.14)*          *+ Contrastive (FID=6.88)*

Figure 8: **Generated samples on CIFAR-100 using UViT-S.** Our proposed methods gradually improve the overall structure, semantics, and details (*e.g.*, see the motorbike, chair and bugs).

## F    LIMITATIONS AND FUTURE RESEARCH DIRECTIONS

Although we conduct extensive experiments to prove our claim that it is possible to enhance both generative and discriminative performance simultaneously through our approaches, our evaluations are limited to datasets up to ImageNet-1k ($256^2$). We did not investigate larger datasets with higher resolutions and larger models, or even on text-to-image generation, due to high computational costs inherently for generative models.

Moreover, some results indicate that achieving optimal performance may rely on data augmentation details, which might require careful tuning. A key remaining challenge for integrating generative

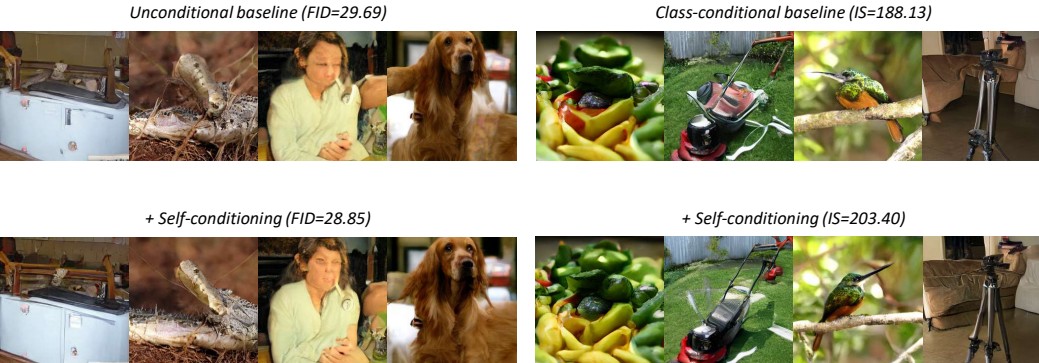

*Unconditional baseline (FID=29.69)*  *Class-conditional baseline (IS=188.13)*

*+ Self-conditioning (FID=28.85)*  *+ Self-conditioning (IS=203.40)*

Figure 9: **Generated samples on ImageNet1k-256x256 using DiT-B.** Our method also improves the overall structure and details in latent-space unconditional (*e.g.*, see the human face and the dog) and class-conditional generation (*e.g.*, see the lawn mower and the tripod).

and discriminative learning, we believe, is the development of effective strategies to organize and identify multiple views of the same instance (even in the latent space), meriting future research.

Finally, though our work provides initial insights into the formation, distribution, and enhancement of internal representations within diffusion models, the precise dynamics governing how these representations evolve throughout the training remain largely underexplored.

