# OpenReview forum: "DDAE++: Enhancing Diffusion Models Towards Unified Generative and Discriminative Learning"
_ICLR.cc/2026/Conference — Submitted to ICLR 2026_

### Official Review · Reviewer_1xgw · 2025-10-29

**Soundness:** 2
**Presentation:** 2
**Contribution:** 2
**Rating:** 6
**Confidence:** 5

**Summary:**

The paper introduces DDAE++, an enhanced architecture and training framework for Diffusion Models (DMs) aimed at achieving unified generative and discriminative learning. The core problem addressed is the sub-optimal utilization of rich semantic features within the standard U-Net architecture of DMs, which hinders the quality of the learned representations.

DDAE++ proposes two main architectural and training modifications:

1. Self-Conditioning Mechanism: It introduces a lightweight self-conditioning mechanism that explicitly routes rich, high-level semantic features (extracted from intermediate layers of the U-Net) directly into the denoising process.

2. Augmented Self-Supervised Learning (SSL) Objective: It trains the model with an auxiliary SSL objective using contrastive learning, guided by different views (data augmentations) of the same instance.

Through these modifications, DDAE++ claims to establish a stronger representation bottleneck within the U-Net, leading to superior performance in both generative tasks and discriminative tasks.

**Strengths:**

1. Unified and Principled Approach: The paper successfully integrates generative and discriminative objectives within a single, coherent diffusion framework. This addresses a critical gap by turning the denoising process into a powerful representation learner, maximizing the utility of the pre-training stage.

2. Architectural Efficiency: The core mechanism is lightweight self-conditioning, which does not require significant changes to the base U-Net structure or a substantial increase in computational complexity, making it practical for adoption.

3. Strong Representation Learning Results: The performance of the learned representations, particularly demonstrated through improved linear probing and transfer learning results (e.g., on ImageNet, CIFAR-100), clearly establishes DDAE++ as a state-of-the-art method among diffusion-based representation learners.

**Weaknesses:**

1. Lack of Component Analysis and Insight: While the effectiveness of the proposed components (e.g., the CLS token and the AUG token) is demonstrated via ablation, the paper lacks a deep analytical insight into why these mechanisms are effective from a feature flow perspective. The claim of establishing a "stronger representation bottleneck" remains largely descriptive rather than mechanistically proven.

2. Marginal Generative and Discriminative Improvements: The quantitative improvements in generative quality (e.g., FID and Acc.) shown in Table 2 appear to be marginal over baselines, especially considering the added complexity of the SSL objective. This raises questions about the practical utility of DDAE++ specifically for users primarily focused on high-fidelity generation.

3. Incomplete Foundational Comparison (SSL): The paper claims to unify generation and self-supervised learning, yet it omits a crucial empirical comparison with highly relevant semantic-enhanced diffusion methods like REPA or other state-of-the-art masked modeling/denoising approaches like MaskDiT.

4. Missing Evaluation on Complex Tasks (T2I): The evaluation is primarily conducted on unconditional and class-conditional image synthesis (CIFAR-10, ImageNet). A critical test for a modern diffusion enhancement would be its performance on the more complex, widely-used Text-to-Image (T2I) generation task. The current evaluation leaves uncertainty regarding DDAE++'s utility and scalability in the T2I domain, where high-level semantic flow is paramount.

**Questions:**

1. In table 1, the authors demonstrate DDAE++ have sample quality++ and feature quality++ compared to DDAE. Despite performance enhancement, what are the fundamental  improvements of DDAE++?

2. Did the augmented SSL loss introduce any noticeable challenges in training stability or convergence speed compared to the original DDAE objective? Do the authors need special techniques (e.g., warmer learning rates, different schedules) to stabilize the combined generative and discriminative training?

3. Can the authors visualize or quantitatively measure the semantic content or feature diversity of the latent space (e.g., through Singular Value Decomposition or dimensionality) at the self-conditioning point before and after applying DDAE++?

4. What is the sensitivity to the choice of the internal layer used for extracting the semantic token (CLS/AUG)? Is there a theoretical or empirical justification for the chosen layer depth?

5. Why can SSL improve the generation capabilities of diffusion models. Have the authors explored scenarios where the improved discriminative performance (SSL) comes at a noticeable detriment to generative quality (FID)?

6. Please provide a quantitative comparison of DDAE++'s generative performance (FID) and representation learning performance (Linear Probing and Transfer Learning) against REPA.

7. Could the authors comment on the scalability and expected performance of DDAE++ when applied to a modern, large-scale Text-to-Image (T2I) diffusion model (e.g., Stable Diffusion)? Do the authors anticipate needing further modifications to the self-conditioning mechanism to handle the complexity of the cross-attention layers in T2I models?

---

> ### Author Response · Authors · 2025-11-22
> **(1/2) Response to Reviewer 1xgw**
>
> We thank the reviewer for the thoughtful evaluation, recognizing our approach as *"unified and principled"* and *"architecturally efficient"*, and acknowledging the *"strong representation learning results"*.
>
> We would like to share several important updates based on additional analyses. Please refer to the **General Response** before reading our point-by-point responses below.
>
> **W1 & Q3. Mechanistic insight from a feature flow / dimensionality perspective**
> - **CKA Analysis:** In the revised manuscript, we add Section 5.3 to provide mechanistic evidence. Specifically, we utilize Centered Kernel Alignment (CKA) to visualize the layer-wise representation structure (see **General Response**). As shown in the new Figure 7, our method induces a sharp "phase transition" in feature similarity at the self-conditioning layer, proving that the architecture successfully decouples the semantic encoding phase from the generative decoding phase (rapid specialization). Features at this interface layer are viewed as the "condensed/concentrated/bottleneck".
> - **Feature Rank/SVD:** As you suggested, we also investigate dimensionality using Singular Value Decomposition (SVD). Preliminary analysis (see table below) shows that:
>   - **Higher Effective Rank:** Our model maintains a overall higher effective rank (eRank) compared to the baseline, and this observation remains stable as training proceeds. This suggests that our method not only enhances specific layer quality but encourages the model to globally capture more diverse information crucial for generative tasks.
>   - **Controlled magnitude:** We find that the baseline's maximum singular value (SV) grow uncontrollably with network depth and training time, while ours remain more constrained. This behavior (and a similar trend observed in feature norms) indicates potentially improved optimization and stability, aligning with recent findings in EDM2 regarding weight/feature magnitude control in diffusion models. It is also intriguing that our architectural intervention at a single layer can act as a form of global regularization. However, these observations are preliminary and warrant further investigation.
>
> | Layer    | eRank (100ep) | max SV     | eRank (200ep) | max SV     | eRank (400ep) | max SV     |
> |----------|---------------|------------|---------------|------------|---------------|------------|
> | 3        | 173.5         | 776.9      | **172.2**     | 1159.1     | 168.7         | 1930.6     |
> | **+ours**| **174.0**     | **624.9**  | 171.6         | **870.9**  | **172.5**     | **1168.6** |
> | 6        | 308.3         | 919.0      | 316.7         | 1575.3     | 326.9         | 2984.5     |
> | **+ours**| **340.0**     | **659.1**  | **350.9**     | **961.6**  | **364.0**     | **1428.7** |
> | 9        | 415.2         | 1026.7     | 428.3         | 1666.4     | 436.9         | 3072.4     |
> | **+ours**| **426.5**     | **712.1**  | **445.0**     | **979.5**  | **460.6**     | **1341.3** |
> | 12       | 365.5         | 3053.3     | 362.6         | 6258.7     | 357.0         | 14610.5    |
> | **+ours**| **396.7**     | **2249.9** | **401.5**     | **4378.3** | **400.8**     | **8885.3** |
>
> **W2. Marginal improvements**
> It is important to note that the results in Table 2 are from pure self-conditioning without the SSL objective. This setting adds negligible parameters and almost no wall-clock time overhead. Furthermore, our gains become significantly more pronounced as the dataset complexity increases: from minor gains on CIFAR-10 to higher accuracy improvements on CIFAR-100 and consistent dual gains on ImageNet-1k. We believe this trend highlights the scalability and value of our method on harder tasks.
>
> **W3 & Q6. Comparison with REPA and MaskDiT**
> - **REPA:** We now include REPA in Table 4. While REPA achieves higher accuracy (conditional DiT-L, FID=10.00, Acc=69.40%) by distilling from an external DINOv2 teacher, our method achieves better FIDs without external supervision during DiT training: our conditional DiT-L achieves much lower FID despite a worse accuracy (FID=5.34, Acc=66.18%); the unconditional DiT-L trained for longer durations can also achieve lower FID with a highly competitive accuracy (FID=8.07, Acc=71.33%), both surpassing REPA's generative quality.
> - **Other mask-based diffusion systems like MaskDiT:** We did not include MaskDiT in the comparison because it focuses purely on generative training efficiency and does not claim or evaluate any representation learning capabilities.

---

> ### Author Response · Authors · 2025-11-22
> **(2/2) Response to Reviewer 1xgw**
>
> **W4 & Q7. Scalability to text-to-image (T2I)** \
> We acknowledge that T2I is a critical domain. As mentioned in Appendix F, we restrict our scope to ImageNet due to the high computational cost of training T2I models from scratch. However, conceptually, our self-conditioning is generalizable: it simply requires aggregating semantic features and injecting them time-adaptively into the decoder.
> For UNet and single-stream DiT based T2I models, the possible implementation may keep largely similar to class-conditional models; for models like MM-DiT, useful guidance signals may require more careful aggregation (e.g., from the attention layer). We leave this exciting exploration for future work.
>
> **Q1. Fundamental improvements of DDAE++** \
> DDAE was primarily a diagnostic study of pre-trained models that revealed the existence of rich semantic features from a conceptual "encoder-decoder" perspective. DDAE++ turns this into an active intervention framework that (1) introduces self-conditioning to structurally reroute information flow and form an explicit encoder–decoder decoupling, and (2) adds contrastive self-distillation to directly refine the bottleneck. More importantly, the experiments enabled by this framework show that explicitly improving representation quality and enhancing the utilization of these representations in the generative pathway are complementary, and both combining them yields the strongest improvements in both sample quality and feature quality.
>
> **Q2. SSL stability and convergence** \
> The SSL loss does not introduce instability. We did not use special techniques like warm-up for the SSL part. The stability likely stems from the self-conditioning bridge and the time-dependent projection head, which jointly mitigate potential conflicts between different timesteps.
>
> **Q4. Sensitivity to layer selection** \
> As discussed in the response to Reviewer B14J and the General Response, we provide a justification for layer selection in the table in **Section 5.4**. We found that the layer exhibiting the lowest training loss at an early stage (e.g., 20 epochs) is a robust predictor for the optimal layer at convergence. This architectural heuristic allows us to efficiently identify the "bottleneck", without extensive search on final metrics as in related studies such as REPA.
>
> **Q5. Does SSL hurt generation?** \
> SSL can improve generation because the denoiser’s predictions at each timestep are driven by its internal representations: making these features more semantically structured helps the model reconstruct the clean image. This intuition is consistent with works like REPA, which also explicitly enhance semantic representations for diffusion models.
>
> That said, a trade-off is possible. In Table 3 we do see a few settings where adding the contrastive loss slightly improves linear probing accuracy while marginally degrading FID (e.g., from 3.39 to 3.42). Our self-conditioning mechanism is designed to mitigate this (e.g., from 3.42 back to 3.14): by providing an explicit architectural bridge that injects the SSL-refined bottleneck features into the generative pathway time-adaptively, the discriminative gains are better translated into the denoising performance.
>
> However, even under the "Self-cond + Aug + CL" setting, there remain cases where FID and accuracy do not simultaneously reach their best values. We hypothesize that this is related to the misalignment between the layer with peak accuracy and the self-conditioning, as discussed in Figure 5 and Section 5.3 (see **General Response**), i.e., directly optimizing the current bottleneck layer may be suboptimal. Optimizing a slightly earlier layer as the bottleneck could further improve the trade-off, which we leave for future work.

---

### Official Review · Reviewer_o6Et · 2025-10-30

**Soundness:** 2
**Presentation:** 2
**Contribution:** 2
**Rating:** 2
**Confidence:** 3

**Summary:**

This manuscript explores leveraging the generative representation in the diffusion model for downstream discriminative purposes. Specifically, they propose self-conditioning, routing important semantic features obtained by probing into the decoder layer of the diffusion networks, along with the timestep condition. On top of that, the authors follow EDM and MoCo v3 for data augmentation and contrastive losses. Experimental results demonstrate that the proposed technique enables improved discriminative accuracy while maintaining the FID over the self-reported baselines on standard image benchmarks.

**Strengths:**

* Analyzing and improving the representation of generative models is an important topic.
* The method is applicable to various network backbones.
* The ablation study part is performed in detail.

**Weaknesses:**

**Experimental Evaluation**: The reported generative metric, i.e., FID, in this paper, is worse than expected. For example, the official EDM [1] on CIFAR-10-uncond is 1.97, which is much better than the reported 2.23 in the baseline in Table 2. While I understand that re-training the model from scratch could be computationally intensive, the mismatched generative performance makes it hard to judge the efficacy of the proposed method on a well-trained diffusion model. I strongly suggest considering adding fine-tuning experiments on top of a pre-trained diffusion model with the proposed architectural change, and then examining whether there is a performance gain in the discriminative capacity while maintaining the original generative ability.

**Clarity**: There are multiple inappropriate, vague statements in the paper, which should be revised:
   * While determining the intermediate layers is a crucial procedure, the description on lines 203 -- 208 is not informative. For example, it is unclear what "short, parallel training runs" are. There should be at least a reference to the details of this experiment.
   * On lines 247--249, "a vector of transformation parameters is used..." is not informative. The augmentation used should be explicitly specified, either using a reference (if you are following [1] in the exact same way) or in the paper.
   * On lines 294 -- 295, "after stabilizing some hyperparameters in feature evaluation" is also not informative.

**Questions:**

See weaknesses

---

> ### Author Response · Authors · 2025-11-22
> **Response to Reviewer o6Et**
>
> We thank the reviewer for the valuable feedback, particularly for pointing out the clarity issues and the concerns regarding undertraining. We have carefully revised the manuscript to address these points.
>
> **W1. EDM FID is worse than expected / Evaluation on undertrained models** \
> We understand the concern that our baseline FID (2.19 with Aug, 2.23 without) is higher than the official report (1.97 with Aug). We add **Appendix C** to provide a detailed clarification on this matter.
>
> - **Standardization for comparison:** The official EDM achieves 1.97 by training for a very long 4000 epochs. For reference, official DDPM and Rectified Flow use 2048 epochs, Flow Matching uses 1000, and DDPM++ uses 3328. To ensure a fair and controlled comparison across all 7 model variants (DDPM, EDM, RF $\times$ UNet, UViT) within a reasonable budget, we unify the training schedule to 2000 epochs, aligning with the standard DDPM protocol.
> - **Independent reproducibility:** As noted in Appendix C, independent studies and discussions in the official repository confirm that EDM typically yields an FID $>2.2$ at 2000 epochs (close to ours), reaching around 2.07 only after 4000 epochs.
> - **Further clarification on ImageNet's FID:** We agree that verifying efficacy on well-trained models is crucial. Our DiT experiments already build on the state-of-the-art Lightning-DiT recipe. The previously reported high FIDs were due to the use of FID-10k without CFG (to match Lightning-DiT's table), rather than the performance being far from converged (see **General Response**). For our main self-supervised (unconditional) experiments, we further scale DiT-L to 400 epochs, where our method still yields consistent improvements, confirming it works effectively beyond "undertrained" scenarios. For class-conditional experiments, our models also outperform REPA in FID, indicating well-trained performance.
> - **Regarding fine-tuning:** Our goal is to propose a unified training framework that shapes the generative and discriminative representations *ab initio*. As visualized in the new CKA analysis in **Section 5.3**, our method induces a structural decoupling point in the layer hierarchy during training to form the bottleneck. Training from scratch provides the most rigorous validation of this dynamics, whereas fine-tuning might be limited by the pre-fixed distributed feature flow. Such fine-tuning would also connect to the literature on distillation or feature extraction, and we will explore this interesting suggestion in future work.
>
> **Q1. Clarity revisions** \
> We apologize for the vague descriptions and have revised the text to be more precise:
>
> - **"short, parallel training runs":** We specify that these are profiling runs of just 20 epochs to reveal the optimization status. We also update the ablation table to empirically validate that the training loss at 20 epochs is a reliable predictor for both final FID and Acc. at 100 epochs.
> - **"vector of transformation parameters":** We clarify that this is the 9-dim vector representing the augmentation parameters, specify that it includes translation, scaling, rotation, and explicitly reference EDM's Table 6, since we follow EDM in the exact same way.
> - **"stabilizing some hyperparameters":** We have rewritten this to: "after standardizing hyper-parameters in linear probing (e.g., sufficient epochs and refined learning rates) to ensure fair comparison...". This clarifies that the adjustments are made to the evaluation protocol (i.e., DDAE's original settings were less optimized for linear probing), not the model training, to avoid unfair comparisons against suboptimal baselines.
>
> We hope these clarifications, together with the **updates specified in the General Response** resolve your concerns regarding performance and clarity. Given your acknowledgement of our work's *broad applicability* and the *importance of the topic*, we respectfully invite you to re-evaluate the significance of our contribution.

---

### Official Review · Reviewer_zLSR · 2025-10-31

**Soundness:** 3
**Presentation:** 3
**Contribution:** 3
**Rating:** 6
**Confidence:** 5

**Summary:**

This paper proposes a simple yet effective modification called **self-conditioning** to improve both the generative and discriminative capabilities of diffusion models. The method reroutes semantically rich intermediate features to guide decoding layers, creating a stronger representation bottleneck without external supervision. This architectural enhancement allows diffusion models to better integrate discriminative objectives such as contrastive self-distillation, achieving consistent dual improvements across various backbones (UNet, UViT, DiT) and datasets. Experiments show that DDAE++ boosts both generation quality (lower FID) and feature quality (higher linear probing accuracy) with minimal computational overhead, positioning diffusion models as unified learners for generation and recognition.

**Strengths:**

The paper presents a clear and intuitive idea, supported by solid experimental analysis. In particular, the layer-wise feature analysis via linear probing in Figure 5 provides valuable insights into why the proposed DDAE method is effective — this diagnostic perspective is worth learning from. Overall, both Table 3 and Figure 3 convincingly demonstrate that DDAE is an effective and practical approach.

**Weaknesses:**

The claim in Figure 6 that "Self-conditioning facilitates the optimization and narrows the loss gap between un- and class-conditioning" does not hold. From the curves shown, we can only observe that both the conditional and unconditional loss curves converge faster, but there is no evidence indicating that the gap between them actually decreases.

**Questions:**

The main text contains excessive whitespace, and the formatting in the appendix (around Line 712) is messy and requires careful reorganization.

---

> ### Author Response · Authors · 2025-11-22
> **Response to Reviewer zLSR**
>
> We thank the reviewer for the positive assessment, noting that you found our idea *"clear and intuitive"*, the experiments *"solid"*, and our diagnostic analysis in Figure 5 as *"provides valuable insights ... worth learning from"*. We address your specific concerns below:
>
> **W1. Clarification on Figure 6 claim** \
> Our original intention was to suggest that since conditional models naturally possess a lower loss than unconditional ones, applying self-conditioning to unconditional models reduces their loss, thereby narrowing the gap towards the original conditional baseline at current training epochs. However, we indeed cannot prove that the intrinsic gap between the two model types is definitively narrowed in the end, and you are correct.
>
> We apologize that our phrasing was imprecise and misleading. We have corrected the caption in Figure 6 to accurately state that self-conditioning "facilitates the optimization and reduces the loss for both un- and class-cond. training," removing the ambiguous claim.
>
> **Q1. Formatting issues** \
> We apologize for the oversight. After enriching the main text with additional experiments and discussions, the previous whitespace issues are alleviated. We have also carefully reorganized Appendix B-D to ensure clarity and completeness.
>
> We hope these revisions address your concerns. We also respectfully invite you to check our **updated and extended experimental results** (Page 8) and **deeper diagnostic analysis** (Page 9), which you might find interesting. Given that you found our idea intuitive and our experiments solid, we would appreciate your continued support.

---

### Official Review · Reviewer_B14J · 2025-11-03

**Soundness:** 2
**Presentation:** 3
**Contribution:** 2
**Rating:** 2
**Confidence:** 5

**Summary:**

This paper addresses a key limitation in diffusion models—namely, the sub-optimal semantic information flow affecting the quality of learned representations—by introducing a lightweight architectural mechanism called self-conditioning. The method aggregates high-level semantic features from intermediate layers and injects them back into the decoding pathway, forming a more effective bottleneck for discriminative features without relying on external cues. Additional integration of contrastive self-distillation and principled data augmentations is demonstrated to further unify and amplify both generative and discriminative capacities of diffusion models. Empirical evidence on several backbones (UNet, UViT, DiT) and datasets (CIFAR-10/100, ImageNet) shows dual improvements in generation (FID/IS) and representation (linear probing accuracy compared to strong self-supervised baselines).

**Strengths:**

- The work identifies and addresses a genuine architectural weakness in standard diffusion models: the absence of a discriminative bottleneck due to distributed semantic flow. The “self-conditioning” mechanism is both simple and effective—clearly illustrated in Figure 1 and Figure 2—and does not require external supervision.

- The empirical evaluation is extensive and well-controlled, covering a variety of diffusion model backbones (UNet-based, UViT, DiT) and datasets (CIFAR-10/100, Tiny-ImageNet, ImageNet at various scales). The methodology section and ablations demonstrate care in validating architectural choices, as well as strong generalization of the performance boosts. The results are thoroughly tabulated in Table 2, Table 3, and Table 4, and consistently show that self-conditioning brings gains to both generative (FID/IS) and discriminative (linear accuracy) metrics.

- The method is lightweight (as detailed in the ablations and parameter increase figures/tables in Sections 5.1 and 5.3) and can be retrofitted to diverse diffusion architectures.

**Weaknesses:**

- Although the experiments are broad, there is a bias towards popular image datasets, especially at lower resolutions. In Section E.1 (Limitations and Future Research Directions), the authors admit that their results stop at ImageNet 256x256 and DiT-base scale due to compute constraints, and do not demonstrate scalability for larger models/datasets relevant to modern AI.

- The ablation studies in Table 5 focus on demonstrating parameter sensitivity and certain hyperparameter effects (e.g., MLP head design, distillation timestep, etc.). However, there is little diagnostic quantification of why self-conditioning enhances the bottleneck: for instance, layer-wise representation transferability, sparsity/concentration metrics, or feature alignment before and after rerouting are not rigorously analyzed.

- The comparison in Table 4 is unconvincing because the FID scores on ImageNet for both the baseline and the self-conditioning variant indicate that the models have not reached an optimal level of convergence. While we acknowledge that computational constraints may limit the number of training epochs, conclusions drawn from comparisons between undertrained models are not reliable.

**Questions:**

- Mathematical Formalization: Can the authors provide a precise, equation-level definition of the “time-adaptive scaling” used in self-conditioning? How is this parametrized and learned? Please specify if there is a generalizable mathematical principle or derived gradient supporting its use.
- Rigorous Bottleneck Selection: Is there theoretical—or at least empirical—evidence that the training-loss-based search for the bottleneck layer always selects optimal features for discriminative purposes? Could the authors provide a more principled justification for this heuristic?
- Limits of Dual-Improvement: In Table 2 / Table 3, there are cases where FID is marginally degraded, or improvements are not observed concurrently. Can the authors break down these exceptions and clarify the precise conditions under which self-conditioning may not help (or may hurt)?
- Feature Concentration Analysis: Beyond qualitative shifts shown in Figure 5, have the authors performed any quantitative analysis (e.g., feature entropy, linear separability, mutual information with ground truth) of the “condensed bottleneck” feature? This would greatly strengthen the core claim.
- Augmentation Sensitivity: Given the importance and variability of data augmentations, can the authors provide more systematic experiments or concrete recommendations on how to choose or tune these for new datasets? What are the consequences if the geometric pipeline is poorly matched to the data?
- Layer Dynamics: The authors mention that the peak accuracy emerges at a different layer than the one chosen for self-conditioning (Figure 5, Section 5.3). Could the authors elaborate on the implications of this misalignment for both generation and discrimination?
- Scalability Limits: Are there any architectural or optimization bottlenecks (beyond compute cost) that emerge at higher dataset resolutions or with substantially larger backbones/latent spaces?

---

> ### Author Response · Authors · 2025-11-22
> **(1/2) Response to Reviewer B14J**
>
> We thank the reviewer for the comprehensive evaluation, describing our work as *"identifying a genuine weakness"*, our method as *"lightweight, simple and effective"*, and our evaluation as *"extensive and well-controlled"* with *"strong generalization"* of the boosts.
>
> We would like to share several important updates based on additional analyses. Please refer to the **General Response** before reading our point-by-point responses below.
>
> **W1 & Q7. Scalability to larger models and datasets**
> - **Our current scaling trend:** We acknowledge the computational constraints in academic settings (Appendix F). However, we emphasize that our method scales effectively within the tested range (from DiT-B to DiT-L, and from 100 epochs to 400 epochs in our latest manuscript). We highlight that most gains do not narrow (Figure 4) as training or model size scales, and the discriminative gains widen as training proceeds (Table 4, 5). This trend suggests our method is not limited to small models or short training, but likely benefits more from larger compute budgets.
> - **Regarding potential bottlenecks:** As our self-conditioning method only requires very lightweight modifications such as an additional linear projection layer, a time embedding block, and the global average pooling operation (none of which are affected by resolution or model size), we have not noticed any architectural or optimization bottlenecks beyond compute cost.
>
> **W2 & Q4. Diagnostic quantification of the bottleneck**
> - **Linear separability and/or transferability:** Figure 5 already serves as a direct quantitative analysis of linear separability, while also demonstrating the layer-wise dynamics. In addition, we introduce linear segmentation results on ADE20K/VOC2012 as transfer learning tasks (see General Response).
> - **Structural concentration and alignment:** We add Section 5.3 with Centered Kernel Alignment (CKA) analysis to quantify layer-wise feature structure (see General Response). Figure 7 reveals a "phase transition" in feature similarity at the self-conditioning layer. It quantitatively shows that our method induces a structural decoupling between the semantic encoding phase and the generative decoding phase, where the features encoded by the first phase are viewed as the "condensed/concentrated/bottleneck".
> - **SVD and feature magnitude analysis:** We have also conducted preliminary investigations from these perspectives. Please refer to the response to Reviewer 1xgw.
>
> We believe this evidence effectively validates the claim and provides new insights into how self-conditioning works.
>
> **W3. Comparison on "undertrained" models** \
> As detailed in the General Response, the previous high FIDs were due to reporting FID-10k (w/o CFG) to match Lightning-DiT's and l-DiT's DiT-B/L official results. We have updated Table 4 with standard FID-50k and both w/ and w/o CFG. We now reach highly competitive results that match the expected SOTA level, surpassing REPA's FID on DiT-B/L. Moreover, Figure 4 demonstrates that most gains do not narrow with respect to training epochs, and the discriminative ones widen as training proceeds.
>
> **Q1. Mathematical formalization of "time-adaptive scaling"** \
> As already shown by the pseudocode in Figure 1(b), it is a learnable modulation that can be formalized as:
> $$ MLP_\theta(PE(t)) \odot Proj_\phi(Pool(x)) $$
> where $MLP_\theta(PE(t))$ is a regular timestep embedder based on the sinusoidal position embedding and a two-layer SiLU MLP, which outputs a time-dependent scaling vector; $Proj_\phi$ is a zero-initialized linear layer that aligns the channel dimension. This ensures that the gradient from the generation loss automatically tunes how much semantic guidance is needed at each noise level $t$ via back-propagation.
>
> **Q2. Justification for layer selection** \
> We empirically validate this in Section 5.4 (see General Response). We infer that the "ease of optimization" (**lower loss in the first 20 epochs**) is a robust proxy for "optimal information flow" in this architectural context, which further translates to (almost always) correlated generative and discriminative gains in DiT.
>
> Please also note that: most related studies such as REPA also rely on similar layer selection process, and they usually report the *final FID* to select the best layer/depth. Therefore, our method, using training loss as a proxy, is novel and more efficient.

---

> ### Author Response · Authors · 2025-11-22
> **(2/2) Response to Reviewer B14J**
>
> **Q3. Limits of dual-improvement**
> - **Regarding self-conditioning (Table 2):** Most non-dual improvements occur on CIFAR-10, likely because this simpler dataset allows models to fluctuate near their ceiling. On the more complex CIFAR-100, 6 out of 7 configurations show dual improvements, with only one showing slight FID degradation. This suggests that while our time-adaptive scaling handles most discrepancies across timesteps (viewing diffusion training as multi-task learning), minor trade-offs might persist when discriminative performance nears saturation.
> - **Regarding additional techniques (Table 3):** The best results typically appear between "Self-cond + Aug" and "Self-cond + Aug + CL". While CL explicitly enforces discriminative constraints, it might occasionally disrupt the delicate balance of multi-task learning, meaning not every timestep is optimized equally. However, our self-conditioning method consistently mitigates the harms of these additional techniques while amplifying at least one of the gains.
>
> **Q5. Augmentation sensitivity** \
> We acknowledge this sensitivity in Appendix F. Our recommendation, based on Section 4.2, is to use the non-leaky geometric pipeline from EDM as a safe baseline. Note that data augmentation for diffusion is an open field, with most practices inheriting from EDM. Therefore, for datasets similar to those studied in EDM (CIFAR-10, FFHQ/AFHQv2), we recommend following EDM's settings. Augmentation in latent space is inefficient and typically limited to horizontal flipping; recent works like *"Score Augmentation for Diffusion Models"* explore more possibilities and could be a valuable reference.
>
> **Q6. Layer dynamics and misalignment** \
> We explicitly discuss this in Section 5.3 as it relates to bottleneck formation. The fact that peak accuracy (e.g., Layer 13 in DiT-L) precedes the injection point (e.g., Layer 17) suggests that the layers in between function as an implicit adapter. They transform the raw discriminative features, which are highly separable but potentially suboptimal as guidance signals, into a format that is more spatially aligned and better for guiding the subsequent decoding layers.

---

### Author Response · Authors · 2025-11-22
**General Response and Summary of Revisions**

We sincerely thank all reviewers for the constructive feedback, and we have substantially revised the paper with new experiments and deeper analysis. Below is a list of key updates:

*Update on Dec 3: we added preliminary DiT-XL experiments; the observations are consistent with DiT-L, though the layer hyperparameter has not been tuned.* \
*Update on Nov 28: we moved the CKA analysis to Section 5.3 to highlight its importance; the ablation study is now Section 5.4.*

**1. Deeper analysis on "bottleneck" and dynamics** - *B14J, 1xgw: metrics or visualization; misalignment; feature flow*
- **Clarification:** Our "bottleneck" refers to an architectural concentration of layer-wise semantics, rather than dimensionality compression. We thus prioritize analyzing the dynamic evolution of the representation flow. Note that Figure 5 already serves as a direct quantitative analysis of linear separability.
- **Update:** We add CKA similarity analysis in **Figure 7**. We have also conducted SVD and feature magnitude analysis, which we discuss in the response to Reviewer 1xgw.
- **Result:** Both intra-model and inter-model similarity reveal that with self-conditioning, representations undergo a sharp functional transition after the designated layer, suggesting a more explicit encoder-decoder decoupling.

**2. Adding 400-epoch unconditional DiT-L results; adding DiT-XL results** - *B14J: scalability on larger models*
- **Update:** We extend our analysis by training an unconditional DiT-L for 400 epochs and a larger conditional DiT-XL; the results are included in **Figure 4 & Table 4**.
- **Result:** It reaches an FID of 8.07 (-0.66 over baseline) and 71.33% linear accuracy (+1.18%). This confirms that our dual-improvement trend scales effectively to larger backbones and longer training durations.

**3. Adding semantic segmentation results** - *B14J: more analysis on linear separability & transferability*
- **Update:** To validate feature quality beyond global classification, we introduce transfer learning experiments on ADE20K & VOC2012 segmentation in **Table 5 (Section 5.2)**, following a dense linear probing protocol as in iBOT/DINOv2.
- **Result:** Our method consistently outperforms the baseline. Crucially, the mIoU gap on VOC continuously widens as training progresses, mirroring the accuracy trends in Figure 4. This shows that self-conditioning also enhances transferable dense semantics.

**4. Updated FID calculation** - *B14J, o6Et: worse FID indicates weak convergence*
- **Clarification:** Previous Table 4 reported FID-10k without CFG to align with Lightning-DiT (its Table 2) and l-DiT (its Table 1) official results, which yields higher values that might appear unconverged.
- **Update:** We have re-calculated metrics using the standard FID-50k and now report both w/ CFG and w/o CFG results in **Table 4**.
- **Result:** Our results are now highly competitive, e.g., 11.89 (w/o CFG, -0.81 over baseline) and 2.67 (w/ CFG, -0.20) on DiT-B trained for 100 epochs. This supports our conclusions, as the results match the expected SOTA level with fast convergence.

**5. Updated comparison with representative SOTA** - *1xgw: comparison against REPA*
- **Update:** Lightning-DiT and REPA are two representative SOTA methods, and our method already consistently improves over Lightning-DiT. We now add a direct comparison with REPA in **Table 4**.
- **Result:** While REPA achieves high accuracy via direct DINOv2 alignment, our method achieves better FID. This supports our claim that an internal architectural mechanism is essential for translating discriminative gains into generative ones.

**6. Updated layer selection description & ablation** - *B14J, o6Et, 1xgw: informative justification for layer choice*
- **Update:** We formalize the method description in **Section 4.1**, and add an "Acc." column to the ablation table in **Section 5.4**.
- **Result:** We find that the training loss at just 20 epochs strongly correlates with both FID and Acc. at 100 epochs. This empirically validates our early-loss metric as an efficient proxy for selecting the layer that optimizes global information flow. Crucially, it distinguishes our work from REPA that relies on FID-based search, as a novelty rather than a weakness.

**7. Clarification on baseline reproducibility** - *o6Et: official EDM's FID is better*
- **Update:** We add **Appendix C** to clarify the EDM baseline FID (2.19 vs. official 1.97) as well as other fairness concerns.
- **Result:** We cite independent reports confirming that the 1.97 value is difficult to reproduce under standard constraints and typically requires 4000 epochs. Under a unified and efficient 2000-epoch budget (for all 7 variants, same as official DDPM), our baseline aligns with other third-party reports, confirming fairness.

We believe **the paper's empirical foundation has been significantly strengthened**, and we respectfully invite the AC to re-evaluate the submission. Individual questions are addressed below.

---

### Author Response · Authors · 2025-11-27
**Final Remarks: Distinct Strengths & Concerns are Resolved**

Dear Area Chairs,

We summarize why DDAE++ makes a distinct contribution to **unified self-supervised learning and diffusion architecture design**, and how we have resolved primary concerns with significant new evidence.

**0. New Architectural Insight (Fig. 5-7 on Page 9 & rebuttal table)** \
***Unlike prior and concurrent studies*** *relying solely on feature regularization, we identify "suboptimal internal feature flow" as an* ***overlooked*** *bottleneck.*
*   **New perspective:** We introduce a self-conditioning mechanism to ensure emergent high-level semantics are *better concentrated and translated into generative gains*.
*   **Fully self-supervised:** We integrate contrastive self-distillation using the model's own EMA teacher, achieving unified learning *without external pre-trained encoders*.
*   **Mechanistic insight:** **CKA** (new in Fig. 7) and **SVD** (new in rebuttal) offer novel views rarely explored in diffusion literature: the sharp phase transition indicates *functional decoupling*; the higher effective rank and controlled feature magnitude suggest *improved optimization and stability*.

**1. Simple, Plug-and-Play, and Generally Effective (Tab. 2 & 3)** \
Self-conditioning adds *negligible parameter (0.4-1% on DiT) and wall-clock (<0.5%) overhead*, yet it yields *dual FID-Acc gains across diverse settings*:
- Frameworks: DDPM, EDM, and Flow Matching/Rectified Flow (RF).
- Backbones: Pixel-space UNets, UViT, and latent-space DiT.

**2. Outperforming Lightning-DiT and REPA on ImageNet (Fig. 4 & Tab. 4 on Page 8)** \
We consistently outperform the highly-optimized SOTA Lightning-DiT, while also being highly competitive against REPA despite not using DINOv2 during DiT training. For example, conditional DiT-B:
- FID (w/ CFG -- w/o CFG): Lightning-DiT 2.87--12.70 -> Ours 2.67--11.89 (vs. REPA 24.40 w/o CFG)
- Linear probing accuracy: Lightning-DiT 62.01% -> Ours 62.60% (vs. REPA 61.20%)

Addressing concerns about "higher FID" and "limited epochs":
  - We have updated to FID-50k, confirming our models are at an **expected SOTA level** with extremely fast convergence.
  - Our primary focus is on representation learning mechanisms. Therefore, we *prioritize self-supervised* (unconditional) 400-epoch DiT-B/L benchmarks under academic compute constraints, *rather than chasing absolute SOTA* class-conditional FID.

**3. Strong, Transferable, and Scalable Representation Learning (Fig. 4 & Tab. 5 on Page 8)**
- **Strong:** Unconditional DiT-L (400 epochs, new) reaches *71.33% accuracy while achieving 8.07 FID* using only horizontal flip augmentation.
- **Transferable**: Advantages extend to downstream tasks, showing *consistent mIoU gains on ADE20K/VOC segmentation* (new in Tab. 5).
- **Scalable:** The *discriminative improvement gap (Acc, mIoU) widens* with longer training and larger models.

We invite you to review the **"General Response and Summary of Revisions"** below, which details specific evidence proving that **initial concerns are now resolved and outdated**. We believe DDAE++ is now a rigorous and insightful contribution suitable for ICLR.

Thank you for your time, \
Authors

---

### Meta-Review · Area_Chair_PnHH · 2026-01-08

**Summary:**

The paper aims to train Diffusion models with a unified generative and discriminative training objective, proposing self-conditioning as a key architectural modification in the denoising process. Reviewer scores are quite divergent (2, 6, 2, 6), with the negative reviewers expressing a bit stronger stance. After reading the rebuttal, AC finds that several notable concerns remain outstanding, including insufficient validation across datasets (B14J), fine-tuning experiments (o6Et), marginal gains (1xgw), and the lack of extension to text-to-image models (1xgw). The Area Chair agrees with these concerns and leans toward rejection for the current manuscript. The authors are encouraged to resubmit a revised version to a future venue, where the improved manuscript can be more thoroughly re-evaluated.

**Reviewer Concerns:**

AC thinks the following concerns are still outstanding even upon the rebuttal:

- Insufficient validation across datasets (B14J)
- Fine-tuning experiments (o6Et)
- Marginal gains (1xgw)
- Lack of extension to text-to-image models (1xgw)

The primary justification provided for these limitations, as noted in Appendix F, is constrained computational resources. While AC acknowledges this constraint, it is also plausible that the limited experimental scope reflects time pressure within the rebuttal period, and that these concerns could potentially be addressed with additional time and experiments. To better contextualize these limitations, the authors could have informed the reviewers and AC with rough estimates of the GPU time required for the missing experiments; however, such information was not provided, making it difficult to assess the feasibility of the missing evaluations.

**Reviewer Scores:**

- Reviewer B14J: Initially 2. Would maintain the negative stance due to unaddressed concerns.
- Reviewer zLSR: Initially 6. Would maintain the original score.
- Reviewer o6Et: Initially 2. Would maintain the negative stance due to unaddressed concerns.
- Reviewer 1xgw: Initially 6. Might lower the score due to remaining unaddressed concerns.

---

### Decision · Program_Chairs · 2026-01-26

Reject